# Nitrous oxide respiration in acidophilic methanotrophs

Samuel Imisi Awala [1,2,9], Joo-Han Gwak [1,9], Yongman Kim [1], Man-Young Jung[3,4,5], Peter F. Dunfield [6], Michael Wagner [7,8] & Sung-Keun Rhee [1] ✉

Aerobic methanotrophic bacteria are considered strict aerobes but are often highly abundant in hypoxic and even anoxic environments. Despite possessing denitrification genes, it remains to be verified whether denitrification contributes to their growth. Here, we show that acidophilic methanotrophs can respire nitrous oxide ($N_2O$) and grow anaerobically on diverse non-methane substrates, including methanol, C-C substrates, and hydrogen. We study two strains that possess $N_2O$ reductase genes: *Methylocella tundrae* T4 and *Methylacidiphilum caldifontis* IT6. We show that $N_2O$ respiration supports growth of *Methylacidiphilum caldifontis* at an extremely acidic pH of 2.0, exceeding the known physiological pH limits for microbial $N_2O$ consumption. *Methylocella tundrae* simultaneously consumes $N_2O$ and $CH_4$ in suboxic conditions, indicating robustness of its $N_2O$ reductase activity in the presence of $O_2$. Furthermore, in $O_2$-limiting conditions, the amount of $CH_4$ oxidized per $O_2$ reduced increases when $N_2O$ is added, indicating that *Methylocella tundrae* can direct more $O_2$ towards methane monooxygenase. Thus, our results demonstrate that some methanotrophs can respire $N_2O$ independently or simultaneously with $O_2$, which may facilitate their growth and survival in dynamic environments. Such metabolic capability enables these bacteria to simultaneously reduce the release of the key greenhouse gases $CO_2$, $CH_4$, and $N_2O$.

Anthropogenic emissions of greenhouse gases (GHGs)—primarily carbon dioxide ($CO_2$), methane ($CH_4$), and nitrous oxide ($N_2O$)—are responsible for a historically rapid increase in Earth's average annual temperature of more than 0.2 °C per decade[1,2]. In addition to achieving net-zero $CO_2$ emissions by 2050, significant reductions in the emissions of other GHGs including $CH_4$ and $N_2O$ are now critically needed. Compared to $CO_2$, the warming effect of $CH_4$ is around 28 to 34 times greater[3,4]. However, its much shorter mean lifetime of approximately 12–13 years[5] provides an additional opportunity to mitigate future climate change. Like $CO_2$, $N_2O$—the third most important GHG—has a long half-life (roughly 120 years) in the atmosphere[6], and its warming potential is about 300 times greater than $CO_2$ over a 100-year time scale[1]. In addition, $N_2O$ is a major cause of ozone depletion in the stratosphere[7,8].

[1]Department of Biological Sciences and Biotechnology, Chungbuk National University, 1 Chungdae-ro, Seowon-Gu, Cheongju 28644, Republic of Korea. [2]Center for Ecology and Environmental Toxicology, Chungbuk National University, 1 Chungdae-Ro, Seowon-Gu, Cheongju 28644, South Korea. [3]Interdisciplinary Graduate Programme in Advance Convergence Technology and Science, Jeju National University, Jeju, Republic of Korea. [4]Department of Science Education, Jeju National University, Jeju, Republic of Korea. [5]Jeju Microbiome Center, Jeju National University, Jeju, Republic of Korea. [6]Department of Biological Sciences, University of Calgary, 2500 University Dr. NW, Calgary, AB T2N 1N4, Canada. [7]Division of Microbial Ecology, Department of Microbiology and Ecosystem Science, Centre for Microbiology and Environmental Systems Science, University of Vienna, Althanstrasse 14, A-1090 Vienna, Austria. [8]Department of Chemistry and Bioscience, Center for Microbial Communities, Aalborg University, Fredrik Bajers Vej 7H, 9220 Aalborg, Denmark. [9]These authors contributed equally: Samuel Imisi Awala, Joo-Han Gwak. ✉e-mail: rhees@chungbuk.ac.kr

Although human activities are by far the most important reason for the unprecedented rise in atmospheric GHGs[9], microbial activities also play a direct role in this rise[10,11]. GHG net accumulation is regulated by the biogeochemical source-sink dynamics of GHGs exchanged between terrestrial, marine, and atmospheric reservoirs[9]. GHG production and consumption in both natural and anthropogenic ecosystems are driven primarily by microbes[10,12]. Methane fluxes in natural environments are controlled by activities of methane-producing (methanogenic) and methane-consuming (methanotrophic) microorganisms. It is estimated that 69% of the atmospheric $CH_4$ budget originates from microbial activities (methanogenesis) while about 50−90% of the produced $CH_4$ is oxidized by methanotrophs before reaching the atmosphere[13,14].

Microbes can oxidize methane under aerobic and anaerobic conditions. Aerobic methanotrophs oxidize methane to methanol by employing either particulate methane monooxygenases (pMMO) or soluble methane monooxygenases (sMMO)[15]. There are two ways in which aerobic methanotrophs use molecular oxygen ($O_2$): as the terminal electron acceptor of aerobic respiration and for methane activation via the methane monooxygenase[15]. Under strictly anoxic conditions, anaerobic methanotrophic microorganisms mitigate $CH_4$ emissions by oxidizing methane with alternative terminal electron acceptors including $NO_3^-$, $Fe^{3+}$, $Mn^{4+}$, $SO_4^{2-}$, and humic acid using reverse methanogenesis pathways[16–19]. Furthermore, intra-aerobic metabolism in the nitrite-dependent anaerobic methane-oxidizing bacterium 'Candidatus Methylomirabilis oxyfera' using pMMO was reported[20].

Interestingly, the genomes of some aerobic methanotrophs encode denitrification enzymes including nitrate ($NO_3^-$), nitrite ($NO_2^-$), nitric oxide (NO), and $N_2O$ reductases[21–24]. Surprisingly, however, none of the methanotroph genomes or MAGs known to date encode a complete set of denitrification genes (Supplementary Dataset 1). Kits and colleagues[21,22] demonstrated that some aerobic methanotrophs can couple $NO_3^-$ and $NO_2^-$ reduction to the oxidation of methane and other electron donors, including methanol, formaldehyde, formate, ethane, ethanol, and ammonia in suboxic conditions. However, whether these aerobic methanotrophs are capable of anaerobic growth with $NO_3^-$ and $NO_2^-$ as terminal electron acceptors remain to be seen.

More than two-thirds of $N_2O$ emissions arise from bacterial and fungal denitrification and nitrification processes in soils[25,26]. $N_2O$ emissions are a major concern in acidic environments due to the high production of $N_2O$ via abiotic reactions and the inhibition of biological $N_2O$ reduction[27,28]. Although multiple sources of $N_2O$ exist[25], there is only one known sink for $N_2O$ in the biosphere—the microbial reduction of $N_2O$ to $N_2$, catalyzed by a copper-dependent enzyme, $N_2O$ reductase ($N_2OR$) encoded by nosZ[29]. The NosZ enzymes found in prokaryotes are phylogenetically classified into two clades: the canonical NosZ (clade I NosZ), found mostly in denitrifiers[30], and the recently described cNosZ (clade II NosZ)[31], which has an additional c-type heme domain at the C terminus, found commonly in non-denitrifiers[31,32]. Thus, bacteria and archaea harboring the nosZ-type genes, in particular those classified as incomplete- or non-denitrifiers because they do not encode the full denitrification pathway, are receiving increasing attention in the search for technologies to combat $N_2O$ emissions[32]. Previous studies have reported the presence of the nosZ gene in the aerobic methanotrophs, Methylocystis sp. SC2 (ref. 23) and Methylocella tundrae[24]. Further genomic analysis from this study suggests that this enzyme is present in some other aerobic methanotrophs, too (Supplementary Dataset 1). Pure culture studies have unequivocally shown that denitrifiers can grow by respiring $N_2O$ (refs. 33,34). Moreover, an electron sink/spill role for $N_2OR$ has been proposed for Gemmatimonas aurantiaca T-27 (ref. 35) without biomass production (i.e., growth). Despite the presence of $N_2OR$ in Methylocystis sp. SC2, its ability to grow in anoxia under $N_2O$-reducing conditions is unverified[36]. Thus, the ability to grow by converting $N_2O$ to $N_2$ has not yet been reported for any of the known aerobic or anaerobic methanotrophs, even with non-methane substrates such as methanol.

Methanotrophs using MMO enzymes are considered to be obligate aerobes. Paradoxically, however, they are often detected at high relative abundance in extremely hypoxic and even anoxic zones of peat bogs, wetlands, rice paddies, forest soils, and geothermal habitats[37,38]. It is therefore critical to investigate the ability of aerobic methanotrophs to use $N_2O$ as the sole terminal electron acceptor for energy conservation and biomass production, a metabolic trait that could allow them to thrive in these anoxic ecosystems. Here, we used a multi-faceted approach to investigate the role of $N_2O$ respiration in defining the physiology and ecology of selected aerobic methanotrophs. Growth experiments demonstrated that the presence of $N_2OR$ in an acidophilic proteobacterial methanotroph, Methylocella tundrae T4, and an extremely acidophilic verrucomicrobial methanotroph, Methylacidiphilum caldifontis IT6, enables these organisms to respire $N_2O$ and to produce biomass while oxidizing a wide variety of electron donors, including methanol, acetol, pyruvate, and hydrogen. In contrast to $N_2O$, respiration of $NO_3^-$ and $NO_2^-$ did not support anaerobic growth of these methanotrophs on C1 substrates. We also demonstrate that Methylocella tundrae T4 can reduce both $O_2$ and $N_2O$ simultaneously, allowing it to oxidize more $CH_4$ and generate more biomass under $O_2$-limiting conditions. Our findings significantly expand the potential ecological niche of aerobic methanotrophs and reveal that some methanotrophic microbial strains could be used to mitigate multiple GHG emissions.

## Results and discussion
### $N_2OR$-encoding genes in aerobic methanotrophs
To identify methanotrophs capable of using $N_2O$ as an alternative electron acceptor, publicly available genomes and metagenome-assembled genomes (MAGs) of methanotrophs were screened for nosZ genes. We found genes encoding $N_2OR$ in genomes and MAGs of methanotrophs from three bacterial phyla: Pseudomonadota, Verrucomicrobiota, and Gemmatimonadota (Supplementary Dataset 1). They were confined to the alphaproteobacterial methanotrophs and absent in gammaproteobacterial methanotrophs in the case of the phylum Pseudomonadota and represented by only two genera, Methylocella and Methylocystis, which also accounted overall for the majority of the methanotroph genomes encoding nosZ. Similarly, nosZ genes were exclusively found in one representative genome in each of the phyla Verrucomicrobiota (represented by the genus Methylacidiphilum) and Gemmatimonadota (represented by the candidate genus 'Methylotropicum'), respectively. Phylogenetic analysis of predicted NosZ protein sequences revealed that those found in Methylocella and Methylocystis are from the clade I NosZ lineage, while those found in Methylacidiphilum and 'Ca. Methylotropicum' are from the clade II NosZ lineage (Fig. 1, Supplementary Fig. 1).

Three Methylocella tundrae strains: T4 (re-sequenced genome), PC1 (ref. 39), and PC4 (ref. 39), have nos gene clusters (NGC) (Fig. 1). These are incorporated into nosRZDFYLX operons in strains PC4 and T4 and a nosZDFYLX operon in strain PC1 (Fig. 1). Strain PC1 has truncated nosZ and missing nosR genes. This is most likely due to its genome being highly fragmented into several small contigs containing missing and truncated genes. The NGC composition and operon arrangement, nosRZDFYLX, were largely similar in the genomes of the six $N_2OR$-containing Methylocystis species (Fig. 1), including Methylocystis sp. SC2 (ref. 23), Methylocystis echinoides LMG27198, three in-house Methylocystis echinoides-like isolates (strains IM2, IM3, and IM4), and a metagenome-assembled genome (MAG) of a Methylocystis sp. AWTPI-1 recovered from a water treatment facility[40]. A notable feature in their NGC organization was the absence of the gene encoding the membrane-anchored copper chaperon, NosL, which is primarily involved in Cu(I) delivery to apo-NosZ[41]. Methanotrophs with pMMO usually possess multiple copper chaperones[42] that may complement

NosL, making it non-essential for NosZ maturation. Altogether, the NGC in these alphaproteobacterial methanotrophs has a similar organization to those of clade I N₂O-reducers (Fig. 1). BLAST results further revealed that the individual *nos* genes in the *Methylocella* and *Methylocystis* strains shared a high degree of similarity to each other and other non-methanotrophic *Alphaproteobacteria* (Supplementary Dataset 2). Also, their NosZ proteins share high homology with proteins annotated as twin-arginine translocation (Tat)-dependent N₂OR (35–89%) and also possess the Tat signal peptide with a characteristic SRRx[F|L] motif[43] found in clade I NosZ[32].

The NGC in the genome of *Methylacidiphilum caldifontis* IT6 (ref. 44), comprises a *nosCZBLDFYC* operon (Fig. 1) but lacks the typical *nosX* and *nosR* found in clade I N₂O-reducers[31,32], involved in NosR maturation[45] and electron transfer to NosZ[46], respectively. Notably, the NGC (IT6_00904–11) was found within the cluster of genes (IT6_00903, IT6_00912–7) encoding alternative complex III (refer to Source Data for annotation information). Both the $aa_3$-type and $cbb_3$-type cytochrome *c* oxidase-encoding genes are also located next to these genes. Genes encoding two *c*-type cytochromes (*nosC*) within the *nos* operon (Fig. 1) could serve electron transport functions[47]. Interestingly, BLAST and synteny analyses of the NGC show that the individual genes are most closely related to genes found in genomes of extremely thermophilic *Hydrogenobacter* species of the phylum *Aquificota* (amino acid identities of 72.41–91.96%) (Supplementary

Dataset 2) with a similar genetic organization (Fig. 1). Strain IT6 NosZ shares high similarities to proteins annotated as Sec-dependent N₂OR (35–89%) with an N-terminal Sec-type signal peptide found in clade II NosZ[31,32]; the highest identities (79–89%) were with NosZ proteins from other *Hydrogenobacter* species. *Hydrogenobacter thermophilus* TK-6, a hydrogen-oxidizing bacterium, can completely denitrify $NO_3^-$ to $N_2$ gas[48], indicating the presence of a functional N₂OR. As a result, *Methylacidiphilum caldifontis* IT6 may also have a functional N₂OR due to the high similarity of its NGC to those of *Hydrogenobacter* species. Although genomes of other *Methylacidiphilum* species, including *Methylacidiphilum fumariolicum*, lacked the gene encoding the N₂OR catalytic subunit, NosZ, some genes encoding Nos accessory proteins were found (Supplementary Dataset 2). Interestingly, the N₂OR genes for *Methylacidiphilum caldifontis* IT6 were found in a genomic island (Supplementary Dataset 3) and were most likely acquired through horizontal gene transfer, which is consistent with its NosZ phylogeny (Fig. 1, Supplementary Fig. 1). This is not surprising since many key metabolic genes in verrucomicrobial methanotrophs, including those encoding the MMO, are believed to have been acquired through horizontal gene transfer[49]. As a result, *Methylacidiphilum fumariolicum* strains might have acquired the NGC before losing the key functional genes but retaining some of the accessory genes. Finally, we found a *nosZBDF* operon in the MAG of the uncultured methanotrophic bacterium '*Ca*. Methylotropicum kingii'[50] that resembles clade II NGC, with

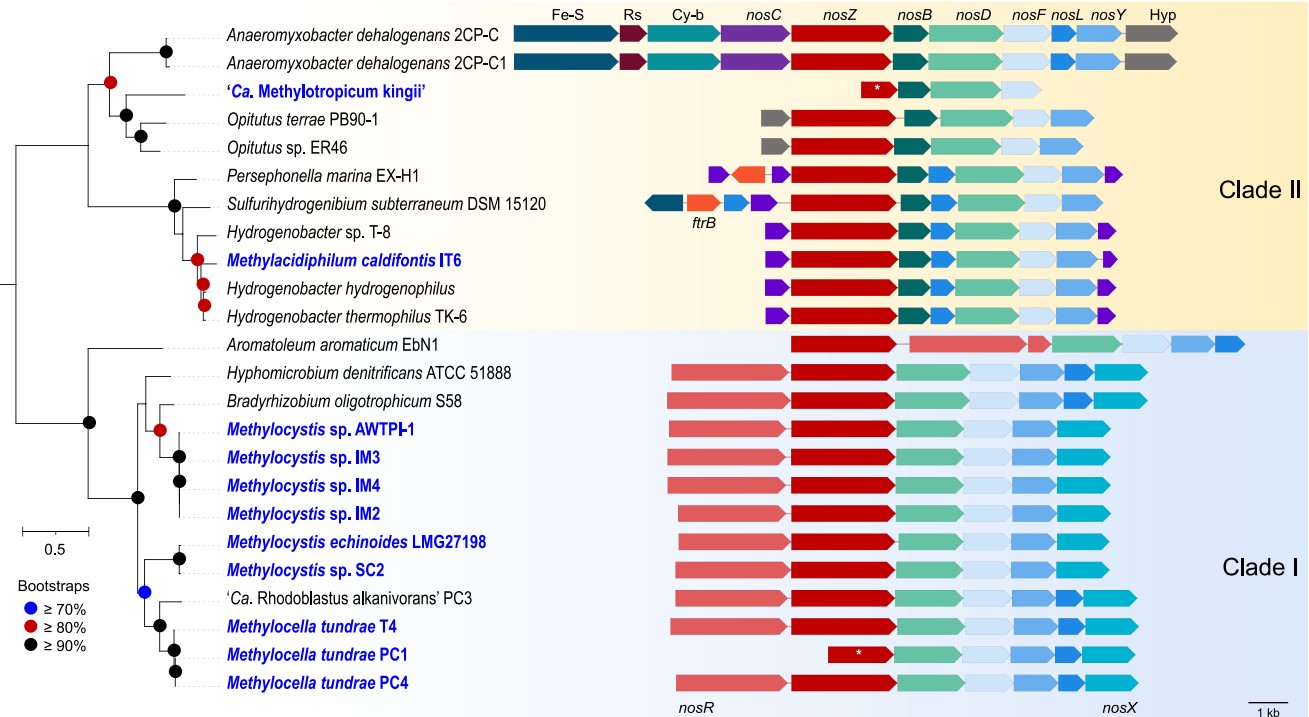

**Fig. 1 | Maximum-likelihood phylogenetic tree of derived NosZ proteins, with *nos* operon arrangements in methanotrophic and non-methanotrophic bacterial strains.** The phylogenetic tree was constructed with IQ-TREE (IQ-TREE options: -B 1000 -m LG + F + R5) using aligned NosZ (details in *Materials and Methods*) and rooted at the mid-point. Bootstrap values ≥ 70% based on 1000 replications are indicated. The scale bar represents a 0.5 change per amino acid position. Organization of the *nos* operon in methanotrophic strains (labeled in blue text) and closely related non-methanotrophic bacteria are shown. The genes, represented by arrows, are drawn to scale. Homologs are depicted in identical colors. The NosZ amino acid sequences and gene arrangement information were retrieved using the following genome accessions: GCF_017310505.1, *Methylacidiphilum caldifontis* IT6; GCF_000010785.1, *Hydrogenobacter thermophilus* TK-6; GCF_011006175.1, *Hydrogenobacter* sp. T-8; GCF_900215655.1, *Hydrogenobacter hydrogenophilus* DSM 2913; GCF_000619805.1, *Sulfurihydrogenibium subterraneum*

DSM 15120; GCF_000021565.1, *Persephonella marina* EX-H1; GCF_000022145.1, *Anaeromyxobacter dehalogenans* 2CP1; GCF_000013385.1, *Anaeromyxobacter dehalogenans* 2CP-C; GCF_003054705.1, *Opitutus* sp. ER46; GCF_000019965.1, *Opitutus terrae* PB90-1; GCF_901905185.1, *Methylocella tundrae* PC4; GCA_901905175.1, *Methylocella tundrae* PC1; CP139089.1, *Methylocella tundrae* T4; FO000002.1, *Methylocystis* sp. SC2; GCF_000025965.1, *Aromatoleum aromaticum* EbN1; GCF_022760775.1, '*Candidatus* Rhodoblastus alkanivorans' PC3; GCF_000143145.1, *Hyphomicrobium denitrificans* ATCC 51888; GCF_000344805.1, *Bradyrhizobium oligotrophicum* S58; GCF_027923385.1, *Methylocystis echinoides* LMG27198; GCA_003963405.1, *Methylocystis* sp. AWTPI-1. * indicates that the *nosZ* genes are truncated due to genome fragmentation. Source Data contains genome annotation information for *Methylocella tundrae* T4, *Methylacidiphilum caldifontis* IT6, *Methylocystis* spp. (strains IM2, IM3, and IM4), and '*Ca*. Methylotropicum kingii'.

a truncated *nosZ* and multiple missing genes like *nosY*, *nosL*, and *nosC* (Fig. 1). These are also likely the result of multiple MAG fragmentations. Multiple sequence alignments of the predicted NosZ proteins of methanotrophs and other microorganisms (clade I and II) were constructed. All the expected metal-binding residues present in $N_2OR$ were mostly conserved in the methanotroph NosZ sequences (Supplementary Fig. 2, Supplementary Note 1).

## $N_2O$-dependent anaerobic growth of methanotrophs

The presence of genes predicted to encode $N_2OR$ in the genomes of *Methylocella tundrae* strains, *Methylacidiphilum caldifontis* IT6, and *Methylocystis* strains (SC2, IM2, IM3, and IM4) (Supplementary Datasets 1, 2) led us to investigate whether this enzyme can support the anaerobic growth of these aerobic methanotrophs when $N_2O$ is supplied as their sole electron acceptor. Physiological studies on $N_2O$ reduction by methanotrophs focused on *Methylocella tundrae* T4 and *Methylacidiphilum caldifontis* IT6 since preliminary experiments showed that the $N_2OR$-containing *Methylocystis* strains, including *Methylocystis* sp. SC2 and the in-house *Methylocystis* strains (IM2, IM3, and IM4) failed to reduce $N_2O$ under various anoxic growth conditions. We set up anoxic batch cultures of *Methylocella tundrae* T4 and *Methylacidiphilum caldifontis* IT6 using methanol as a sole electron donor with or without $N_2O$ as the sole electron acceptor. For these incubations, 2 mM ammonium ($NH_4^+$) was used as the nitrogen source instead of $NO_3^-$ to avoid the involvement of dissimilatory nitrate reduction particularly in the *Methylocella* strains with nitrate-reducing potential. As a negative control, closely related methanotrophs lacking a predicted $N_2OR$ (*Methylocella silvestris* BL2 and *Methylacidiphilum infernorum* IT5, respectively) were included in the study design. The growth experiments were conducted in LSM medium at pH 5.5 for *Methylocella* species (strains T4 and BL2) and at pH 2.0 for *Methylacidiphilum* species (strains IT5 and IT6). As expected, in control incubations provided with $O_2$ as the terminal electron acceptor, all four strains grew on $CH_3OH$ (Fig. 2A, D, G, J). In these controls, the maximum specific growth rates ($\mu_{max}$) of the *Methylocella* strains (strain T4: $\mu_{max} = 2.83 \pm 0.03 \, d^{-1}$; strain BL2: $\mu_{max} = 1.79 \pm 0.05 \, d^{-1}$) were higher than those of the *Methylacidiphilum* strains (strain IT6: $\mu_{max} = 1.57 \pm 0.04 \, d^{-1}$; strain IT5: $\mu_{max} = 1.49 \pm 0.01 \, d^{-1}$).

Under $N_2O$-containing anoxic conditions, *Methylocella tundrae* T4 and *Methylacidiphilum caldifontis* IT6 reduced $N_2O$ and grew on methanol (Fig. 2B, H). When $N_2O$ was depleted, the growth of strains T4 and IT6 ceased. To verify that $OD_{600}$ measurements indicated anaerobic cell growth rather than an artifact such as exopolysaccharide production, we demonstrated that cell counts and counts of 16S rRNA genes increased in parallel with $OD_{600}$ during anaerobic growth (Supplementary Fig. 3). No growth was observed in $N_2O$-free anoxic conditions used as negative controls (Fig. 2C, I). These results demonstrate that the anaerobic growth of these methanotrophs was dependent on $N_2O$ as the sole electron acceptor. The observed $N_2O$ reduction was catalyzed by a functional respiratory $N_2OR$, as the $N_2OR$-lacking relatives (*Methylacidiphilum infernorum* IT5 and *Methylocella silvestris* BL2) used as negative controls did not grow or reduce $N_2O$ under anoxic conditions (Fig. 2E, F, K, L). In addition, other known electron donors of *Methylocella tundrae* T4 and *Methylacidiphilum caldifontis* IT6, which support their aerobic growth[44,51,52], also supported their growth under anoxic $N_2O$-reducing conditions (Supplementary Dataset 4). *Methylocella tundrae* T4 grew on pyruvate and acetol, while *Methylacidiphilum caldifontis* IT6 grew on acetol under anoxic $N_2O$-reducing conditions. Further, molecular hydrogen supported the chemolithoautotrophic growth of *Methylacidiphilum caldifontis* IT6 as the sole electron donor under anoxic $N_2O$-reducing conditions (Supplementary Fig. 4). The transcriptomic analysis (see below) suggests that the group 1d [NiFe] hydrogenase encoded in the genome of *Methylacidiphilum caldifontis* IT6 could be involved in chemolithoautotrophic growth under anoxic $N_2O$ respiring conditions.

*Methylocella tundrae* T4 exhibited a higher growth rate ($\mu_{max} = 0.47 \pm 0.02 \, d^{-1}$) than *Methylacidiphilum caldifontis* IT6 ($\mu_{max} = 0.18 \pm 0.01 \, d^{-1}$) on methanol and $N_2O$. However, these values are approximately 6 and 9 times, respectively, lower than the growth rates measured for both strains under $O_2$-respiring conditions. Biomass yields $Y_{x/m}$ (g DW·mol$^{-1}$ $N_2O$ or $O_2$ reduced) for the methanol-oxidizing cultures of strains T4 and IT6 reducing $N_2O$ as the sole electron acceptor were also lower than for cells reducing $O_2$ as the sole electron acceptor. The biomass yield of *Methylocella tundrae* T4 cells grown anaerobically on $N_2O$ ($4.64 \pm 0.04$ g DW·mol$^{-1}$ $N_2O$ reduced) was approximately 45% of that of aerobically grown cells ($10.41 \pm 0.04$ g DW·mol$^{-1}$ $O_2$ reduced). Similarly, *Methylacidiphilum caldifontis* IT6 had a biomass yield when grown anoxically on $N_2O$ ($2.36 \pm 0.04$ g DW·mol$^{-1}$ $N_2O$ reduced), which was only about 38% of that achieved by aerobically grown cells ($6.27 \pm 0.14$ g DW·mol$^{-1}$ $O_2$ reduced). This improved molar yield on $O_2$ is expected despite the higher reduction potential of $N_2O$ (see Eqs. [1] and [2]), since $O_2$ respiration accepts twice as many electrons as $N_2O$ respiration (Eq. 1 and 2)[53]. In addition, the aerobic terminal oxidases of both strains are proton pumps and conserve energy (Supplementary Datasets 5, 6)[54,55], whereas $N_2OR$ does neither[56]. To our knowledge, our results constitute the first report of $N_2O$ reduction coupled with anaerobic growth in any methanotroph.

$$N_2O + 2H^+ + 2e^- \rightarrow N_2 + H_2O \quad E_{0'}(pH7.0) = +1.36V \quad (1)$$

$$O_2 + 4H^+ + 4e^- \rightarrow 2H_2O + H_2O \quad E_{0'}(pH7.0) = +0.82V \quad (2)$$

It is well known that $N_2O$ reduction is generally inhibited at acidic pH (<6.0)[57], resulting in $N_2O$ accumulation in acidic environments[28,58]. However, the current study revealed that two acidophilic methanotrophs, *Methylocella tundrae* T4 and *Methylacidiphilum caldifontis* IT6 can reduce $N_2O$ in moderately acidic (pH 5.5) and extremely acidic (pH 2.0) conditions, respectively. The existence of acid-tolerant $N_2O$ reducers (pH 4.0 to 6.0) has been proposed in soil microcosm and enrichment experiments[59,60]. So far, the only isolate implicated in $N_2O$ reduction at an acidic pH (5.7) is *Rhodanobacter* sp. C01 isolated from acidic soil in Norway[61]. Our study reveals that $N_2O$ reduction can occur even at an extremely acidic pH of 2.0. Furthermore, the conditions required for $N_2O$ reduction in the $N_2OR$-containing *Methylocystis* strains remain unresolved. Perhaps some unknown growth or environmental factors are required to stimulate $N_2O$ respiration in these methanotrophs, which will require further investigation.

## Nitrate and nitrite reduction in *Methylocella* species

**No anoxic growth of *Methylocella* species with $CH_3OH$ and $NO_3^-$.** We next tested if the presence of denitrification enzymes in *Methylocella tundrae* T4 (nitrate reductase [NAR], nitric oxide reductase [NOR] and $N_2OR$) and *Methylocella silvestris* BL2 (NAR, nitrite reductase [NIR], and NOR) (Supplementary Dataset 1) can equate to growth when $NO_3^-$ or $NO_2^-$ is used as the sole terminal electron acceptor. Indeed, the presence of NAR (and NIR) in these methanotrophs resulted in $NO_3^-$ (and $NO_2^-$) reduction when methanol was provided as the sole electron donor. However, growth was barely detected under these conditions (Fig. 3A, B). Strain T4, which lacks a canonical NIR, reduced all the provided $NO_3^-$ stoichiometrically to $NO_2^-$ when provided with methanol as the sole electron donor (Fig. 3A). Under the same condition, strain BL2, a NAR and NIR-containing methanotroph, initially reduced the provided $NO_3^-$ to $NO_2^-$ and eventually, all the accumulated $NO_2^-$ was stoichiometrically reduced to $N_2O$ towards the end of the incubation (Fig. 3B). These results demonstrate that these methanotrophs have a functional NAR and or NIR and can utilize $NO_3^-$ and/or $NO_2^-$ instead of $O_2$ as a terminal electron acceptor. Nevertheless, these methanotrophs do not appear to rely on these activities for growth.

Likewise, other aerobic methanotrophs have demonstrated denitrification activities under suboxic conditions. For example, the gammaproteobacterial methanotrophs *Methylomonas denitrificans* FJG1 and *Methylomicrobium album* BG8 were discovered to couple the oxidation of diverse electron donors to $NO_3^-$ and $NO_2^-$ reduction, respectively[21,22]. However, none of these strains was demonstrated to couple this activity to growth, prompting us to investigate the possible reasons behind the lack of growth (see below). It should be noted that the genomes of all known *Methylacidiphilum* strains lack genes encoding a respiratory NAR (Supplementary Dataset 1).

**Toxicity of reactive nitrogen species for *Methylocella* species.** Considering that methanol oxidation was coupled to $N_2O$ reduction

and led to obvious growth in the $N_2OR$-containing methanotrophs (Figs 2B, H), the lack of growth during $NO_3^-$ reduction by these microorganisms is suspected to be caused by the accumulation of growth-arresting reactive nitrogen species (RNS) like $NO_2^-$ and NO (refs. 62,63). Consistent with this hypothesis, the accumulation of $NO_2^-$ in suboxic cultures of *Vibrio cholerae* and other bacterial species was found to limit population expansion but nitrate reduction still promoted cell viability[64]. $NO_2^-$ typically accumulates due to a lack of functional NIR as observed for strain T4 (Fig. 3A) and, to some degree, even transiently accumulates in the presence of a functional NIR, as observed for strain BL2 (Fig. 3B). The impact of $NO_2^-$ accumulated from $NO_3^-$ reduction might be more severe in acidic environments since protonation of $NO_2^-$ leads to the formation of free nitrous acid

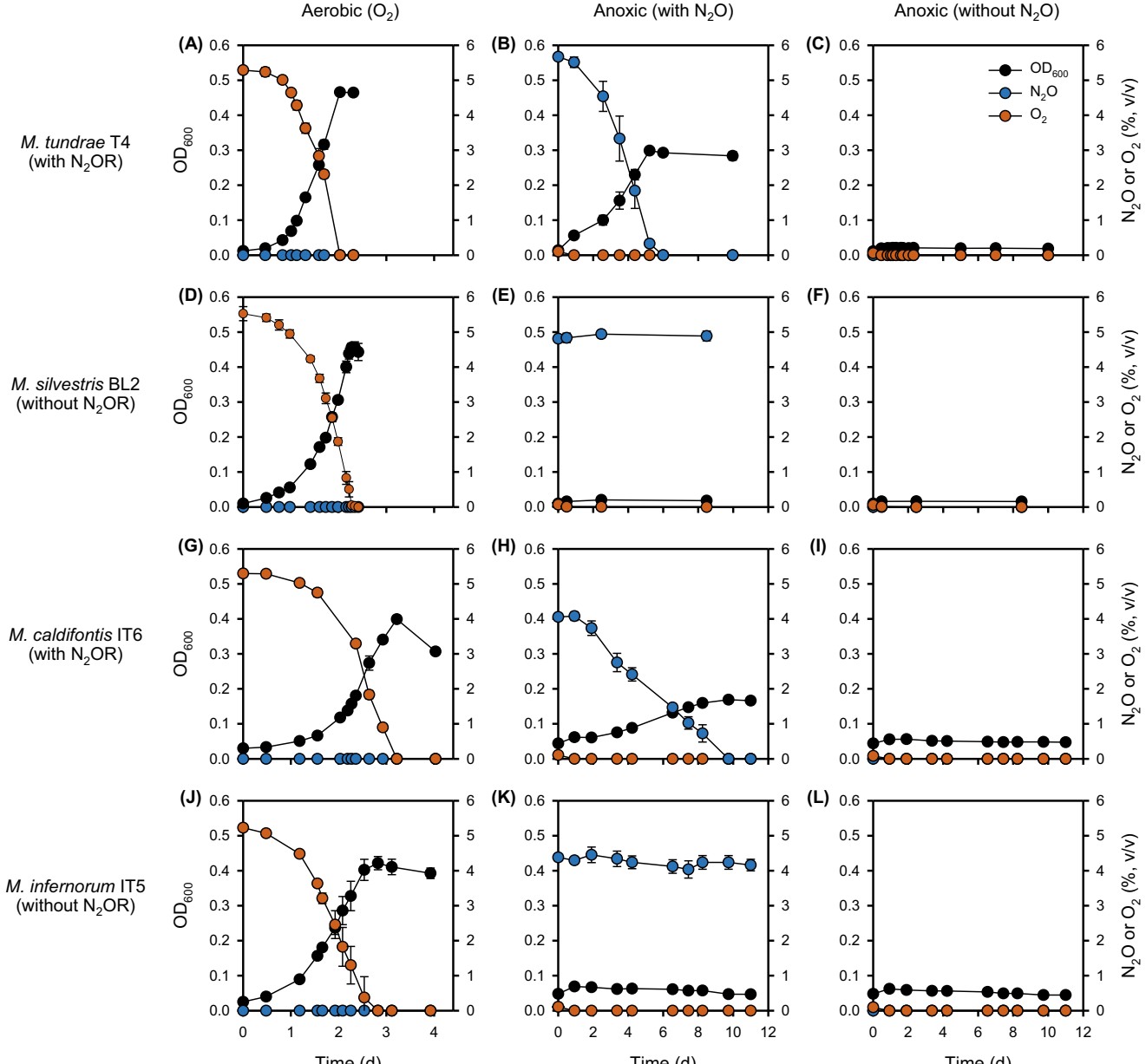

**Fig. 2 | Aerobic and anaerobic growth of $N_2OR$-containing and $N_2OR$-lacking *Methylocella* and *Methylacidiphilum* strains on methanol.** *Methylocella tundrae* T4, *Methylocella silvestris* BL2, *Methylacidiphilum caldifontis* IT6, and *Methylacidiphilum infernorum* IT5 cells were grown in LSM medium supplemented with 30 mM methanol as the electron donor and $NH_4^+$ as the N-source. Aerobic growth of the 4 strains with $O_2$ (**A**, **D**, **G**, **J**), anaerobic growth with $N_2O$ (**B**, **E**, **H**, **K**), and anaerobic growth without $N_2O$ (**C**, **F**, **I**, **L**) as the sole terminal electron acceptor were determined by optical density measurements at 600 nm, followed by measurements of $O_2$ and $N_2O$ consumption in the headspaces of the culture bottles. Note that the trace $O_2$ present at the start of the incubation in the anaerobic cultures without $N_2O$ did not contribute to obvious growth (**C**, **F**, **I**, **L**). All experiments were performed in triplicates. Data are presented as mean ±1 standard deviation (SD), and the error bars are hidden when they are smaller than the width of the symbols. Source data are provided as Source Data file.

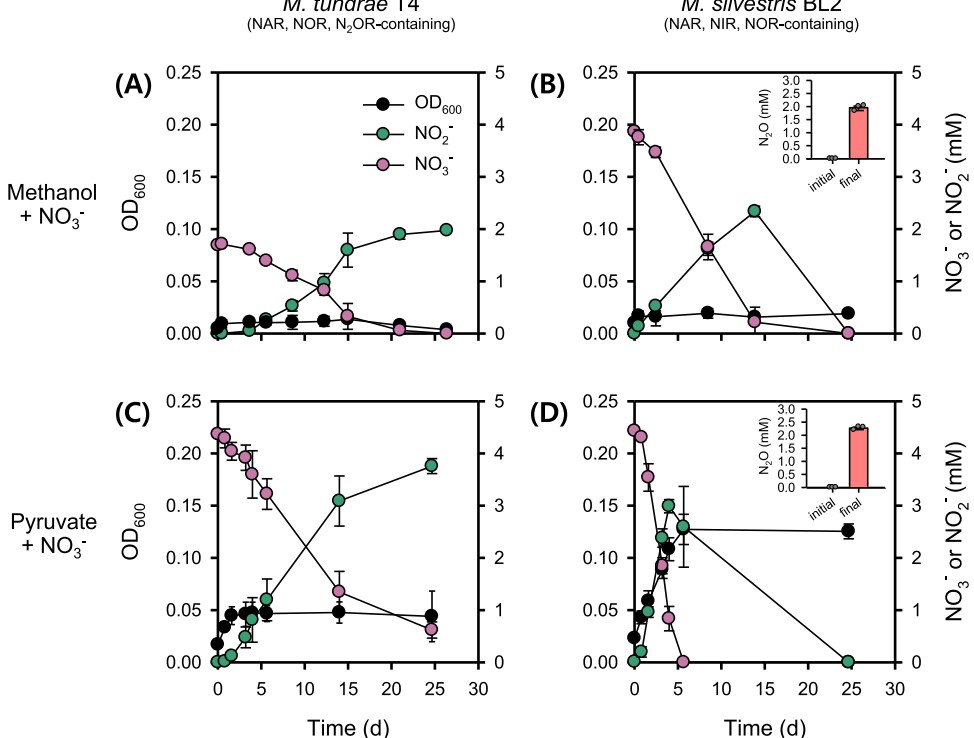

**Fig. 3 | Anaerobic growth of *Methylocella* strains on methanol or pyruvate as the terminal electron acceptor.** *Methylocella tundrae* T4 and *Methylocella silvestris* BL2 cells were grown in LSM medium supplemented with 30 mM methanol and 2–4 mM $NO_3^-$. $NH_4^+$ (2 mM) was supplied as the N-source. Anaerobic growth of *Methylocella tundrae* T4 (**A**) and *Methylocella silvestris* BL2 (**B**) cells on methanol as the sole electron donor with $NO_3^-$ as the sole electron acceptor. Anaerobic growth of *Methylocella tundrae* T4 (**C**) and *Methylocella silvestris* BL2 (**D**) cells on pyruvate as the sole electron donor with $NO_3^-$ as the sole electron acceptor. $N_2O$ produced from $NO_3^-$ reduction by cells of *Methylocella* *silvestris* BL2 grown on methanol or pyruvate is shown as an inset plot within each figure. $N_2O$ production was not observed in strain T4, hence inset plots for $N_2O$ production were not displayed. Lower $NO_3^-$ (ca. 2.0 mM) was used in the case of methanol (**A**) to avoid $NO_2^-$ toxicity. Growth was determined by optical density measurements at 600 nm, followed by measurements of $NO_3^-$ and $NO_2^-$ concentrations. Data are presented as mean ± 1 SD of triplicate experiments, and the error bars are hidden when they are smaller than the width of the symbols. Source data are provided as Source Data file.

(FNA), a known inhibitor of microbial anabolic and catabolic processes[65]. In addition, chemodenitrification of $NO_2^-$ (ref. 66) could result in an accumulation of NO in the cell environment, which is highly toxic to microbial life[67]. To further support the hypothesis of RNS toxicity, strain T4 was cultivated under $N_2O$-reducing conditions with methanol as the sole electron donor and supplied with $NO_3^-$ instead of $NH_4^+$ as the N source in the medium (Supplementary Fig. 5). Consistent with the idea that $NO_2^-$ accumulation results in growth arrest, the culture growth plateaued at approximately the same time $NO_2^-$ accumulated (≥ 0.3 mM $NO_2^-$) (Supplementary Fig. 5A), whereas in control cultures containing $NH_4^+$ instead of $NO_3^-$ as the N-source, $NO_2^-$ accumulation was not observed, and the cells were able to reach higher cell densities (Supplementary Fig. 5B). Furthermore, the effect of $NO_2^-$ stress induced in strain T4 was verified by adding varying $NO_2^-$ concentrations (0, 0.01, 0.03, 0.1, 0.3, and 1 mM) to aerobic (Supplementary Fig. 6A) and anaerobic $N_2O$-respiring cultures (Supplementary Fig. 6B). Nitrite, particularly at concentrations higher than 0.3 mM at pH 5.5, induced stress in *Methylocella tundrae* T4, resulting in growth inhibition (Supplementary Fig. 6). These results are comparable to that of *Methylophaga nitratireducenticrescens* JAM1, a facultative methylotroph, which, when grown aerobically on methanol at pH 7.4, had a four-fold decrease in biomass in the presence of 0.36 mM $NO_2^-$ and did not grow in the presence of 0.71 mM $NO_2^-$ (ref. 68). Taken together, our data suggest that the failure of $NO_3^-$/$NO_2^-$-reducing methanotrophs to grow on methanol may result from RNS toxicity. On the other hand, when $N_2O$ is reduced to $N_2$ by $N_2O$-reducing methanotrophs, the creation of these RNS is avoided, which may explain the disparity in growth with $N_2O$ as the terminal electron acceptor compared to $NO_3^-$ and $NO_2^-$.

**Toxicity of C1 metabolites in nitrate-reducing *Methylocella* species.**
Aside from the inhibitory effects of RNS, toxic intermediates from methanol metabolism might synergistically contribute to the inability of methanotrophs to grow when respiring $NO_3^-$/$NO_2^-$. Although formaldehyde is a key intermediate in the C1 metabolic pathway in many methylotrophs, it is highly toxic[69]. Therefore, in situations where biomass production is limited due to RNS toxicity, it is likely that formaldehyde further retards the growth of denitrifying methanotrophs. To investigate this mechanism, we grew *Methylocella* strains under $NO_3^-$-reducing conditions using a C-C electron donor, pyruvate, which does not generate formaldehyde as a major metabolite (Figs. 3C, 3D). Eventually, nearly all the supplied $NO_3^-$ was stoichiometrically converted to $NO_2^-$ and $N_2O$ in strains T4 and BL2, respectively. In contrast to the lack of growth on methanol, pyruvate supported the growth of both *Methylocella* strains under $NO_3^-$-reducing conditions (Fig. 3C, D). Growth was more pronounced in strain BL2 than in strain T4 (Fig. 3C, D), possibly due to the presence of NIR and NOR in addition to NAR in strain BL2, which limited $NO_2^-$ accumulation (Fig. 3D). Nonetheless, no further growth on pyruvate was observed in strain BL2 after day 5, despite reduction of the accumulated $NO_2^-$ (~2.5 mM) to $N_2O$ (Fig. 3D). It is worth noting that the accumulated $NO_2^-$ concentration (Fig. 3D) is higher than the 0.3 mM concentration that inhibited *Methylocella tundrae* T4 (Supplementary Fig. 6) and may also be responsible for the lack of growth in strain BL2.

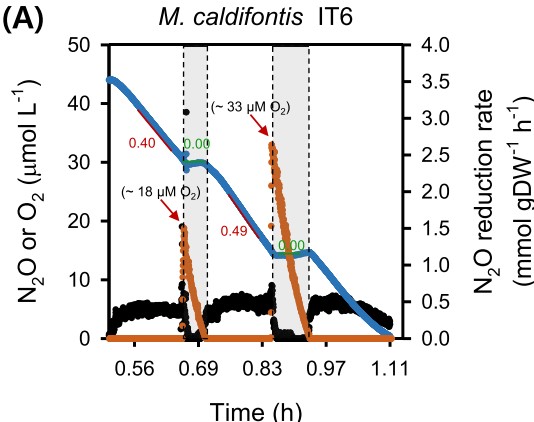

**Fig. 4 | Microrespirometry-based $N_2O$ and $O_2$ reduction during methanol oxidation by N2OR-containing methanotrophs.** $N_2O$ and $O_2$ reduction by cells of *Methylacidiphilum caldifontis* IT6 (**A**) and *Methylocella tundrae* T4 (**B**) during methanol oxidation. Filled blue dots represent dissolved $N_2O$, filled orange dots represent dissolved $O_2$, and filled black dots represent $N_2O$ reduction rates.

Experiments were performed in a microrespirometry (MR) chamber fitted with $O_2$ and $N_2O$ microsensors. The red arrows mark the addition of 14–33 μM $O_2$ into the MR chamber. The red- and green-marked numbers close to the red and green lines represent the $N_2O$ reduction rates before and during $O_2$ reduction (gray-shaded area) in the MR chamber, respectively. Source data are provided as Source Data file.

Overall, these results demonstrate that in the tested *Methylocella* strains: (i) RNS have a major inhibitory effect on growth under denitrifying conditions; (ii) there are no growth benefits from methanol oxidation coupled to $NO_3^-$ reduction, probably due to toxic C1 metabolic intermediates as well as RNS; and (iii) anaerobic growth is observed when $NO_3^-$ reduction is coupled to the oxidation of pyruvate, a C-C electron donor; although the amount of growth is dependent on the completeness of the denitrification pathway and the accumulation of RNS. These propositions are supported by increased expression of genes involved in RNS and C1 metabolite detoxification under denitrifying conditions (see transcriptomic analysis below). Taken together, these results may explain why methanotrophs that couple methanol oxidation to $NO_3^-$ or $NO_2^-$ reduction show no clear signs of growth due to this process. Most methanotrophs can only utilize methane and its C1 derivatives as energy sources[70] and thus should not be able to grow under denitrifying conditions[21,22]. On the other hand, versatile facultative methanotrophs of the genus *Methylocella* are potentially able to grow in strictly anoxic habitats when alternative multi-carbon substrates are available. In terrestrial environments, various nitrogen oxides, originating from nitrification and denitrification processes, coexist and are spatiotemporally dynamic[71]. Thus, depending on the versatility of $NO_2^-$ and NO reduction potential of methanotrophs as well as their coexistence with other $NO_2^-$ and NO-reducing microorganisms, $N_2O$ respiration can be supported or compromised (see Supplementary Figs 5, 6).

## N₂O reduction coupled with CH₃OH or CH₄ oxidation

**N₂O reduction kinetics.** We investigated $N_2O$ respiration kinetics using resting cells of anaerobic $N_2O$-respiring cultures ($CH_3OH + N_2O$) in a microrespirometry (MR) chamber. Harvested cells of strains *Methylacidiphilum caldifontis* IT6 and *Methylocella tundrae* T4 were dispensed into a closed 10-mL MR chamber outfitted with $O_2$ and $N_2O$-detecting microsensors, supplied with $CH_3OH$ (2 mM) and $N_2O$ as a sole electron donor and acceptor, respectively, and incubated anoxically. The $N_2O$ respiration kinetics followed Michaelis-Menten kinetics (Supplementary Fig. 7, Supplementary Note 2). The cells of strains T4 and IT6 grown at anoxic $CH_3OH + N_2O$ conditions reduced $N_2O$ at a maximum rate of $1.122 \pm 0.005$ mmol $N_2O \cdot h^{-1} \cdot g$ $DW^{-1}$ (Supplementary Fig. 7A) and $0.414 \pm 0.003$ mmol $N_2O \cdot h^{-1} \cdot g$ $DW^{-1}$ (Supplementary Fig. 7B), respectively. The molar ratios of $CH_3OH$ to $O_2$ and $CH_3OH$ to $N_2O$ consumed were approximately 1:1.0 ($\pm 0.05$; $n = 3$) and 1:2.04 ($\pm 0.17$; $n = 3$), respectively, which coincide with the theoretical

values obtained from Eqs. 3 and 4.

$$CH_3OH + O_2 \rightarrow 0.5CO_2 + 1.5H_2O + 0.5CH_2O(biomass) \quad (3)$$

$$CH_3OH + 2N_2O \rightarrow 0.5CO_2 + 1.5H_2O + 2N_2 + 0.5CH_2O(biomass) \quad (4)$$

**Sensitivity of N₂OR to O₂.** While $O_2$ is well known to impair N2OR activity[72], some bacterial strains have been reported to reduce $N_2O$ in the presence of $O_2$ (refs. 73,74). We therefore tested the capacity of strains IT6 and T4 to reduce $N_2O$ in the presence of $O_2$ by using resting cells of anoxic $CH_3OH + N_2O$ cultures. After spiking $O_2$ to strain IT6 cells respiring $N_2O$ in the anoxic MR chamber, $N_2O$-respiration ceased: dropping from the maximum (0.4–0.5 mmol $N_2O \cdot h^{-1} \cdot g$ $DW^{-1}$) to zero (Fig. 4A, Table 1). $N_2O$ reduction activity only started when the dissolved $O_2$ concentration was below ca. 3 μM, suggesting the $N_2O$ reduction activity of this strain is highly sensitive to $O_2$. In contrast, when $O_2$ (~14 and 30 μM) was added to $N_2O$-respiring cells of strain T4, simultaneous reduction of $N_2O$ and $O_2$ was observed (Fig. 4B). However, the $N_2O$ respiration rates dropped to 0.24 and 0.13 mmol $N_2O \cdot h^{-1} \cdot g$ $DW^{-1}$ after spiking ~14 and 30 μM $O_2$, respectively, which were approximately 34 and 20% of the maximum rate before $O_2$ introduction (0.64–0.68 mmol $N_2O \cdot h^{-1} \cdot g$ $DW^{-1}$). These results suggest that in contrast to strain IT6, $N_2O$ reduction in strain T4 is not highly impaired by $O_2$. N2OR activity fully recovered in both strains after $O_2$ was depleted. Because the N2OR of strain IT6 was found to be highly sensitive to $O_2$, further characterization of methanotroph N2OR activity in response to $O_2$ exposure was limited to strain T4.

Considering these results, we set out to see if cells of strain T4 could continue $N_2O$ respiration while using $O_2$ for $CH_4$ oxidation in the MR chamber. The cells used for this experiment were cultured in suboxic conditions with starting gas mixing ratios (v/v) of 1% $O_2$, 5% $N_2O$, and 20% $CH_4$ (i.e., $CH_4 + O_2 + N_2O$ condition). Similar to the anoxic $CH_3OH + N_2O$-adapted cells described above, the suboxic $CH_4 + O_2 + N_2O$-adapted cells co-respired $O_2$ and $N_2O$ after injecting $CH_4$ (~406 μM) into a 5-mL MR chamber containing $O_2$ (~30 μM) and $N_2O$ (~480 μM) (Fig. 5A). Interestingly, the maximum $N_2O$ respiration rates during each $O_2$ spike were 1.4 to 2 times higher (1.58–2.47 mmol $N_2O \cdot h^{-1} \cdot g$ $DW^{-1}$) in the suboxic $CH_4 + O_2 + N_2O$-adapted cells (Fig. 5B, Table 1) than in the anoxic $CH_3OH + N_2O$-adapted cells ($1.12 \pm 0.01$ mmol $N_2O \cdot h^{-1} \cdot g$ $DW^{-1}$) (Table 1, Supplementary Fig. 7B), suggesting that the cells can modulate the rates of $N_2O$ reduction in response to $O_2$ availability.

**Table 1 | Microrespirometry-based substrate-specific N₂O- or O₂-reduction rate by *Methylocella tundrae* T4 cells grown under anoxic and suboxic growth conditions**

| Condition | Rate (mmol·h⁻¹·g DW⁻¹) |
|---|---|
| Maximum respiration rates of anoxic CH₃OH + N₂O-respiring cells | |
| N₂O respiration (at 0 μM O₂; electron donor CH₃OH; at the first 4 spikes of N₂O) | 1.12 ± 0.01 |
| N₂O respiration (at 0–5 μM O₂; electron donor CH₃OH; at the 5th spikes of N₂O) | 0.64–0.68 |
| N₂O respiration (at O₂ > 5 μM; electron donor CH₃OH) | 0.13–0.24 |
| O₂ respiration (at O₂ > 5 μM; electron donor CH₃OH) | 1.02–1.07 |
| Maximum respiration rates of suboxic CH₄ + N₂O + O₂-respiring cells | |
| N₂O respiration (at 25–60 μM O₂; electron donor = CH₄) | 1.58–2.47 |
| N₂O respiration (at 5–170 μM O₂; electron donor = CH₄) | 1.32 ± 0.25 |
| O₂ respiration (at 25–60 μM O₂; electron donor = CH₄) | 0.98–2.37 |
| O₂ respiration (at 5–170 μM O₂; electron donor = CH₄) | 0.95 ± 0.09 |

These values were obtained from respiration activities with cells that had CH₃OH or CH₄ as the sole electron donor.

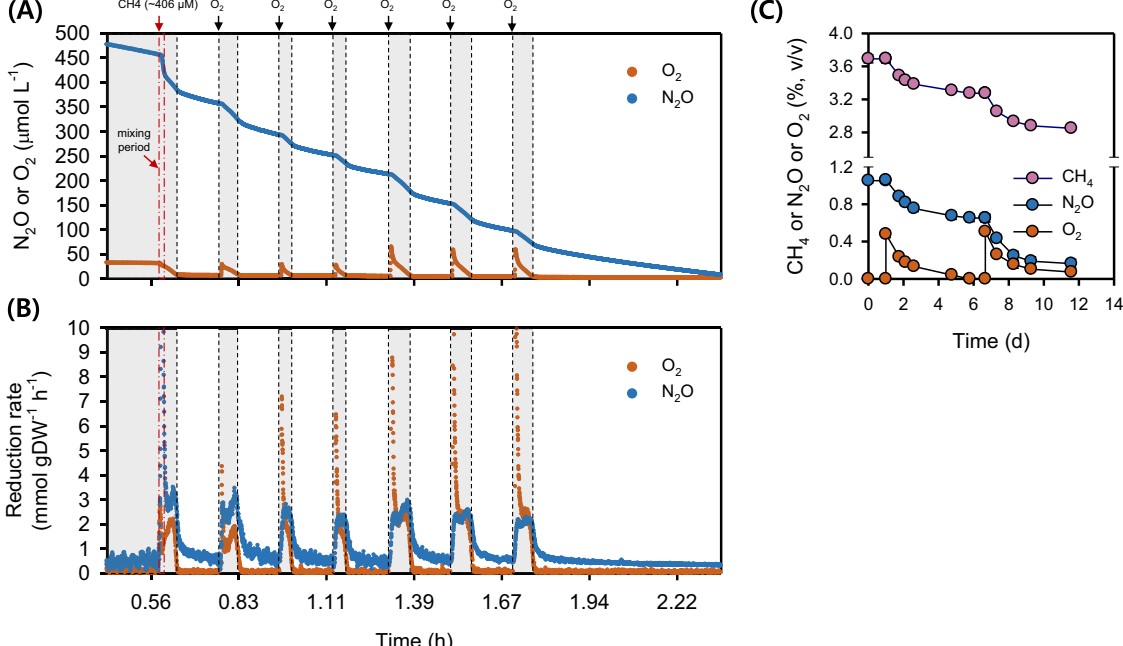

**Fig. 5 | Simultaneous N₂O and O₂ reduction by *Methylocella tundrae* T4 cells during CH₄ oxidation in microrespirometry (MR) and growth experiments. A** MR experiment showing the simultaneous reduction of N₂O and O₂ by *Methylocella tundrae* T4 cells during CH₄ oxidation. **B** N₂O and O₂ reduction rates by cells of strain T4 during CH₄ oxidation calculated from (**A**). The filled orange and blue dots in the upper (**A**) represent the concentrations of dissolved O₂ and N₂O, respectively. The filled orange and blue dots in the bottom (**B**) represent the rates of O₂ and N₂O reduction, respectively. Experiments were performed in a MR chamber fitted with O₂ and N₂O microsensors. The red arrow marks the addition of CH₄ (~406 μM) into the MR chamber. The black arrow marks the addition of ~26 μM or ~60 μM O₂ into the MR chamber. The gray-shaded area represents points where N₂O and O₂ are reduced simultaneously. **C** Growth experiment showing *Methylocella tundrae* T4 cells reducing N₂O and O₂ simultaneously during CH₄ oxidation. The culture was grown in 2-liter sealed bottles (triplicates) containing 60 mL of LSM medium with 2 mM NH₄⁺ as the N-source. The headspace of the bottles was composed of CH₄ (5%, v/v), O₂ (0.5%, v/v), N₂O (1.4%, v/v), and CO₂ (5%, v/v) and supplemented with additional O₂ (~0.5%, v/v) before its depletion. The incubation period shown in (**C**) is after the initial 20-day incubation period. After the depletion of O₂, additional O₂ was spiked to observe the simultaneous reduction of O₂ and N₂O during CH₄ oxidation. Data are presented as the mean ± 1 SD of a triplicate experiment, and the error bars are hidden when they are smaller than the width of the symbols. Source data are provided as Source Data file.

Accordingly, the maximum rates of N₂O reduction (1.58–2.47 mmol N₂O·h⁻¹·g DW⁻¹) and O₂ reduction (0.98–2.37 mmol O₂·h⁻¹·g DW⁻¹) by the suboxic CH₄ + O₂ + N₂O-adapted cells were comparable (Fig. 5B, Table 1). As the O₂ concentration and reduction rate decreased, the N₂O reduction rate also decreased (Fig. 5A, B), revealing that activation of CH₄ by O₂ is required for stimulating N₂O respiration by CH₄ + O₂ + N₂O-adapted cells. Based on these results, we conclude that, under suboxic conditions, both aerobic CH₄ oxidation and N₂O reduction were operating in concert: O₂ was needed for the

monooxygenase, but the N₂OR remained active and was able to accept electrons released downstream in the C1 oxidation pathway. This adds to the evidence that aerobic N₂O respiration occurs in strain T4 and is linked to aerobic CH₄ oxidation.

Finally, we estimated the O₂ concentration range at which the suboxic CH₄ + O₂ + N₂O-adapted cells of strain T4 show N₂O-reducing activity. At a O₂ concentration of 170 μM, O₂ and N₂O were reduced simultaneously (Supplementary Fig. 8A, B). The maximum N₂O reduction rate (Table 1) was nearly constant (1.32 ± 0.25 mmol

**Table 2 | The effect of N$_2$O addition on CH$_4$-oxidizing cultures of *Methylocella tundrae* T4 growing in suboxic conditions**

| Culture condition | CH$_4$ oxidized (mmol·L$^{-1}$) | O$_2$ reduced (mmol·L$^{-1}$) | N$_2$O reduced (mmol·L$^{-1}$) | Increase in OD$_{600}$ |
|---|---|---|---|---|
| CH$_4$ + O$_2$ | 9.74 ± 0.39 | 14.93 ± 0.43 | NA | 0.114 ± 0.006 |
| CH$_4$ + O$_2$ + N$_2$O | 12.19 ± 0.24 | 13.96 ± 0.41 | 10.15 ± 0.35 | 0.143 ± 0.002 |

The experiment was performed in 2-liter sealed bottles (replicates) with 60 mL of LSM medium in an O$_2$-limiting suboxic headspace with and without N$_2$O (0.5% O$_2$, 5% CH$_4$, 5% CO$_2$, and 0 or 1% N$_2$O). Following the observation of N$_2$O reduction in bottles containing N$_2$O, the headspace O$_2$ and N$_2$O mixing ratios in the bottles were increased to approximately 1% and 2% (v/v), respectively. The reduction of N$_2$O by the cultures increased CH$_4$ oxidation and biomass compared to cultures containing only O$_2$. Data are presented as mean ± 1 SD (*n* = 3). NA not available.

N$_2$O·h$^{-1}$·g DW$^{-1}$) across the O$_2$ concentration range of 5–170 μM (Supplementary Fig. 8B, C) and was about 1.4 times higher than the maximum O$_2$ reduction rates (0.95 ± 0.09 mmol O$_2$·h$^{-1}$·g DW$^{-1}$). This means that even when exposed to high levels of O$_2$, the N$_2$OR in the suboxic CH$_4$ + O$_2$ + N$_2$O-adapted cells remained functional and could reduce N$_2$O at high rates. Other bacterial strains' N$_2$OR activities have been reported at O$_2$ concentrations between 100 and 260 μM (refs. 73,74), indicating that their N$_2$OR activity is similarly O$_2$-tolerant[73] as that of strain T4. According to the findings of Wang and colleagues[73], N$_2$O reducers with an O$_2$ tolerant N$_2$OR maintain low internal O$_2$ concentrations in their cells by rapidly consuming O$_2$, allowing the N$_2$OR to remain active. However, it remains unclear if *Methylocella tundrae* T4 employs a similar strategy to maintain an O$_2$-tolerant N$_2$OR.

**Improved methanotrophic growth of *Methylocella tundrae* in the presence of N$_2$O.** Based on the MR experiments showing the simultaneous reduction of O$_2$ and N$_2$O by CH$_4$-fed cells of strain T4, alongside the clear N$_2$O-dependent anaerobic growth, we hypothesized that strain T4 growth can be enhanced when it oxidizes CH$_4$ by simultaneously reducing O$_2$ and N$_2$O under suboxic conditions. Using fed-batch growth experiments, we verified that strain T4 grows by CH$_4$ oxidation coupled with co-respiration of N$_2$O and O$_2$ (Fig. 5C, Table 2), strongly supporting the MR results above. Cells grown under the suboxic CH$_4$ + O$_2$ + N$_2$O condition consumed roughly the same amount of O$_2$ and N$_2$O (Fig. 5C, Table 2), and these values were comparable to what CH$_4$ + O$_2$ + N$_2$O-grown cells consumed in the MR experiments (see Fig. 5A). Consequently, our results demonstrate that in an O$_2$-limited environment, the cells can benefit energetically by directing more O$_2$ to the monooxygenase step of CH$_4$ oxidation, and simultaneously running a hybrid (O$_2$ + N$_2$O) electron transport system as shown in Table 2 and Fig. 5C.

The data showed unequivocally that the total electron equivalents released during CH$_4$ oxidation to CO$_2$ could account for the total electron acceptor (O$_2$ + N$_2$O) reduced. Based on a CH$_4$ to O$_2$ ratio of 1:1.57 (ref. 75), the total amount of O$_2$ reduced (13.96 mmol·L$^{-1}$) by the suboxic CH$_4$ + O$_2$ + N$_2$O cultures could theoretically only account for 8.89 mmol·L$^{-1}$ oxidized CH$_4$. However, a larger total of 12.19 mmol·L$^{-1}$ CH$_4$ was oxidized by this culture (Table 2), and the excess 3.29 mmol·L$^{-1}$ must have required an additional electron acceptor (i.e., N$_2$O). Consistently, about 10.15 mmol·L$^{-1}$ N$_2$O was reduced by the suboxic CH$_4$ + O$_2$ + N$_2$O cells, equivalent to 5.08 mmol·L$^{-1}$ O$_2$, since half as many electrons are consumed per mol during N$_2$O reduction to N$_2$ compared to O$_2$ reduction to H$_2$O. By running the N$_2$O respiration system, the cells lower their O$_2$-demand for respiration by the aerobic terminal oxidase and maximize O$_2$ use by the methane monooxygenase[76]. Due to having more CH$_4$ oxidized per O$_2$ reduced (~37%) when N$_2$O is present, higher cell densities (OD$_{600}$) per O$_2$ reduced (~34%) were reached in the suboxic CH$_4$ + N$_2$O + O$_2$ cultures than in the O$_2$-replete CH$_4$ + O$_2$ cultures (Table 2), further demonstrating the beneficial contribution of N$_2$O reduction to growth on CH$_4$ at suboxic conditions.

## Transcriptomics

The overall regulation of key genes involved in denitrification and methane oxidation is depicted in Fig. 6 as well as in the supplementary material (Supplementary Figs. 9, 10, 11, Supplementary Datasets 5, 6, 7). Differences in expression were considered significant if the Log$_2$FC was higher than [0.85] or lower than [-1.0] with an adjusted $p \le 0.05$.

**N$_2$OR (O$_2$ replete vs. anoxic conditions).** The transcript levels of the N$_2$OR-encoding genes (T4_03941–7), *nosRZDFYLX*, were 2- to 4.7-fold higher in strain T4 cells respiring N$_2$O in the anoxic CH$_3$OH + N$_2$O conditions compared to strain T4 cells respiring O$_2$ in the O$_2$-replete CH$_3$OH + O$_2$ conditions (Fig. 6 Supplementary Datasets 5, 6). Cells of strain IT6 respiring N$_2$O in the anoxic CH$_3$OH + N$_2$O conditions showed transcriptional upregulation (1.9–6.7-fold) of four *nos* genes (*nosC1BZC2*) under anoxic conditions (Supplementary Fig. 10, Supplementary Dataset 7). Other *nos* operon genes (*nosYFDL*; IT6_00904–11) were expressed constitutively under both the anoxic CH$_3$OH + N$_2$O and O$_2$-replete CH$_3$OH + O$_2$ conditions. The NosC1 and NosC2 proteins of *Wolinella succinogenes* were predicted to facilitate electron transfer from menaquinol to the periplasmic NosZ during the reduction of N$_2$O to N$_2$ (ref. 47) and are likely to play a similar role in strain IT6. Although the exact function of NosB has yet to be elucidated, Hein and colleagues[77] used a non-polar *nosB* deletion mutant of *Wolinella succinogenes* to show that it is necessary for N$_2$O respiration. Overall, increased expression of N$_2$OR-encoding genes in *Methylocella tundrae* T4 and *Methylacidiphilum caldifontis* IT6 cells during anaerobic growth indicates that the N$_2$OR is functional in these methanotrophs and supports their ability to respire and grow using N$_2$O as a terminal electron acceptor.

**N$_2$OR (O$_2$ replete vs. suboxic conditions).** Transcript levels of N$_2$OR-encoding genes were 2–10.7-fold higher in strain T4 cells grown under suboxic CH$_4$ + O$_2$ + N$_2$O conditions than in cells grown under O$_2$-replete CH$_4$ + O$_2$ conditions (Supplementary Fig. 9, Supplementary Datasets 5, 6). This finding is consistent with the N$_2$O respiration activity and growth of strain T4 under suboxic CH$_4$ + O$_2$ + N$_2$O conditions (Fig. 5), in which the cells can efficiently oxidize more CH$_4$ (see Table 2), most likely because the use of N$_2$O for cellular respiration allows them to devote more O$_2$ to CH$_4$ oxygenation.

**Methanol dehydrogenase (O$_2$ replete vs. anoxic conditions).** In methanotrophs, methanol oxidation occurs in the periplasmic space by PQQ (pyrroloquinoline quinone)-dependent methanol dehydrogenase (MDH). Seven PQQ-dependent alcohol dehydrogenases (ADHs)[78] are encoded in the genome of strain T4 (Supplementary Dataset 5). Five are type I ADHs (quinoproteins), which include one calcium-dependent MDH (MxaF-type MDH), and four lanthanide-dependent MDHs (XoxF-type MDH), divided into clades 1 (XoxF1), 3 (XoxF3), and 5 (XoxF5; 2 copies) (Supplementary Fig. 12). The other two are type II ADHs (quinohemoproteins). In addition to the PQQ-dependent ADH, *Methylocella tundrae* T4 and *Methylacidiphilum caldifontis* IT6 genomes contain genes encoding cytosolic Zn$^{2+}$-dependent ADH, which are part of a large family of enzymes that oxidize alcohols to aldehydes or ketones and reduce NAD(P)$^+$ or a similar cofactor[79] (Supplementary Datasets 5, 6).

Among the four XoxF-type MDHs encoded in the genome of strain T4, genes in a *xoxFGJ* operon (T4_03519–21), which include a gene encoding a XoxF5 enzyme, were found to be constitutively transcribed at high levels in cells grown under both O$_2$-replete CH$_3$OH + O$_2$ and

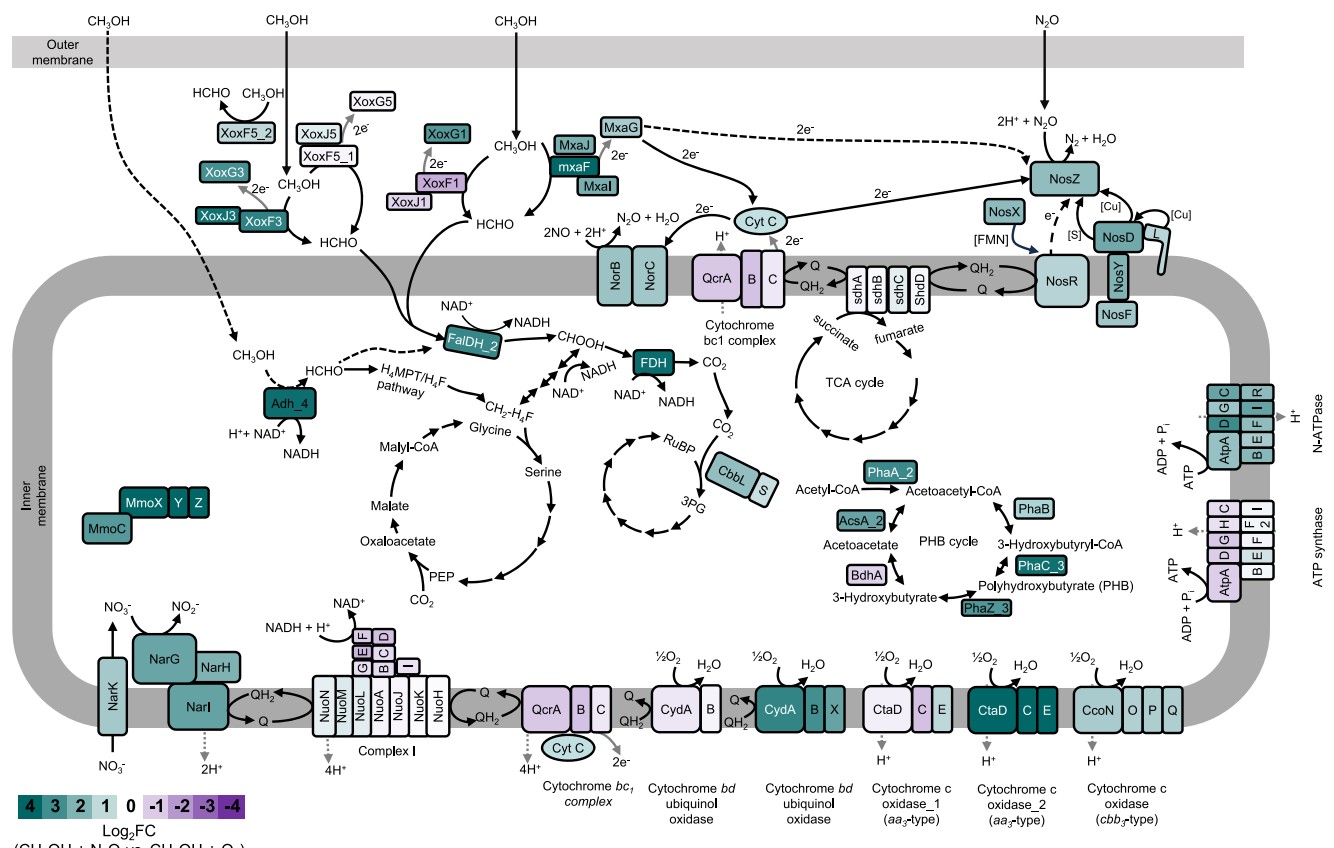

**Fig. 6 | Metabolic reconstruction and transcriptional response of *Methylocella tundrae* T4 cells to O₂-replete (CH₃OH + O₂) and anoxic (CH₃OH + N₂O) methanol-oxidizing growth conditions.** The genes used to reconstruct the metabolic pathway are listed in Table S5. The gene products are shaded according to the relative fold change (Log₂FC) in gene expression between cells grown under anoxic (CH₃OH + N₂O) and O₂-replete (CH₃OH + O₂) conditions. Genes up-regulated in CH₃OH + N₂O-grown cells are shown in teal green, while those up-regulated in CH₃OH + O₂-grown cells are shown in purple. Note that proteins are not drawn to scale. Methanol oxidation: Methanol is oxidized to formaldehyde in the periplasmic space by the PQQ-dependent methanol dehydrogenase (Xox- and Mxa-type), T4_03519–21, T4_00353–5, and T4_01862–76. The NAD(P)⁺-dependent alcohol dehydrogenase (T4_03199) may also be involved in methanol oxidation to

formaldehyde in the cytoplasmic space during anaerobic growth on methanol. Formaldehyde oxidation to formate then proceeds via the tetra-hydromethanopterin (H₄MPT) pathway, and C1 incorporation into the serine cycle is mediated by the tetrahydrofolate (H₄F) carbon assimilation pathway. The Calvin-Benson-Bassham pathway is also a possible route for CO₂ fixation. Nitrous oxide reduction: N₂O is reduced to N₂ through the activity of nitrous oxide reductase in the periplasmic space. Electron transfer to NosZ occurs via cytochrome c from the cytochrome bc1 (Qcr) complex[136,137]. Electron transfer to the NosZ may also involve direct interaction with methanol dehydrogenase C-type cytochrome (XoxG, MxaG). The NosR protein may be involved in the transfer of electrons to NosZ (refs. [136,137]).

---

anoxic CH₃OH + N₂O conditions (Fig. 6, Supplementary Datasets 5, 6). Thus, the *xoxF5* gene likely encodes the predominant MDH used by strain T4 in both O₂-respiring and N₂O-respiring cells. The other sin-gleton *xoxF5* gene (T4_03691) and a *xoxF3* gene found in a separate *xoxFGJ* cluster (T4_00353–5) were also significantly upregulated in cells grown under anoxic CH₃OH + N₂O conditions in comparison to cells grown under O₂-replete CH₃OH + O₂ conditions (Fig. 6, Supplementary Datasets 5, 6). Furthermore, we observed a significant upre-gulation (2- to 22-fold) of the genes encoding MxaFI-type MDH (T4_01872–6) in the anoxic CH₃OH + N₂O-grown cells (Fig. 6, Supplementary Datasets 5, 6). Thus, our results indicate the use of various MDHs by strain T4 during anaerobic growth. In strain IT6 a *xoxF* gene encoding a XoxF2-type MDH is present as part of the *xoxGJF* operon (IT6_00336–8) (Supplementary Dataset 7) and the expression of the *xoxF2* gene was 2-fold upregulated in the N₂O-respiring cells (Supplementary Fig. 10, Supplementary Dataset 7).

A cytosolic Zn²⁺-dependent ADH bound to NAD(P)⁺ is known to perform the oxidation of methanol in Gram-positive methylotrophs[80]. A Zn²⁺-dependent ADH (T4_03199) of strain T4 was significantly upregulated (13.8-fold) in the anoxic CH₃OH + N₂O-grown cells com-pared to the O₂-replete CH₃OH + O₂-grown cells (Fig. 6, Supplementary

Datasets 5, 6). Strain IT6 genome also contained three copies of genes encoding enzymes annotated as Zn²⁺-dependent ADH (Supplementary Dataset 7). The expression of two of these genes (IT6_01501 and IT6_01931) were 3.9-fold and 2.5-fold upregulated in N₂O-respiring cells compared to cells respiring O₂ (Supplementary Fig. 10, Supple-mentary Dataset 7). Even though PQQ-dependent MDHs have a high-affinity for and activity with methanol as a substrate, their use in strictly anoxic conditions will be limited because PQQ biosynthesis requires molecular oxygen[81]. Thus, PQQ-dependent MDHs are sug-gested to be functional at completely anoxic conditions only when PQQ is carried over from an aerobic growth stage or provided externally[82]. On the other hand, Zn²⁺-dependent MDHs have the advantage of utilizing a ubiquitous cofactor, NAD(P)⁺, and can be functional during anaerobic growth[83]. This finding raises the possibility that strains T4 and IT6 can employ alternative ADHs such as the Zn²⁺-dependent ADH to facilitate methanol oxidation in strict anoxia. Some genes required for the subsequent steps of C1 metabolism, i.e., for-maldehyde and formate dehydrogenases, were also upregulated in strain T4 (but not IT6) growing anaerobically. These are depicted in Fig. 6 and supplementary materials (Supplementary Fig. 9, Supple-mentary Dataset 5).

**Methanol dehydrogenase (O$_2$ replete vs. suboxic conditions).** Furthermore, we also examined expression levels of genes encoding MDHs in strain T4 cells grown under suboxic CH$_4$ + O$_2$ + N$_2$O conditions (Supplementary Fig. 9, Supplementary Datasets 5, 6). Genes in the cluster T4_01862–76, which encodes the calcium-dependent MDH (MxaF-type MDH), had the highest transcript expression among all MDH-encoding genes in CH$_4$-oxidizing cells grown under suboxic CH$_4$ + O$_2$ + N$_2$O conditions. When compared to O$_2$-replete CH$_4$ + O$_2$ conditions, the expression of genes within this cluster was 1.8- to 371.5-fold upregulated (Supplementary Fig. 9, Supplementary Datasets 5, 6). This is unexpected since genes encoding the Mxa-type MDH are typically downregulated in the presence of lanthanides[84]; which we also included (2 μM each of cerium and lanthanum) in the growth medium. Their apparent upregulation (even when lanthanides are present) suggests that this enzyme might play an important role in CH$_4$ metabolism in the presence of N$_2$O and suboxic conditions. As observed above, genes in the *xoxFGJ* operon (T4_03519–21) were also highly expressed at the suboxic conditions (Supplementary Fig. 9, Supplementary Datasets 5, 6), suggesting that this key MDH is used by strain T4 in all three conditions. Genes in the cluster T4_01892–4 including the gene encoding the XoxF1 MDH were also significantly upregulated (18- to -35-fold) in the suboxic CH$_4$ + O$_2$ + N$_2$O-grown cells compared to the O$_2$-replete CH$_4$ + O$_2$-grown cells. The operon T4_02097–8, which encodes a cytochrome *c*550 (T4_02097) and a type II ADH (T4_02098), exhibited 6-fold and 30.6-fold upregulation, respectively, in cells grown under suboxic CH$_4$ + O$_2$ + N$_2$O conditions as opposed to cells grown under O$_2$-replete CH$_4$ + O$_2$ conditions. In addition, two Zn$^{2+}$-dependent ADHs (T4_03097 and T4_03199) were significantly upregulated (3.5-fold and 46-fold, respectively) in strain T4 cells grown under suboxic CH$_4$ + O$_2$ + N$_2$O conditions compared to cells grown under O$_2$-replete CH$_4$ + O$_2$ conditions. Overall, it appears that cells oxidizing methanol under anoxia (CH$_3$OH + N$_2$O-grown cells) or those oxidizing methane under suboxia (CH$_4$ + O$_2$ + N$_2$O-grown cells) use a distinct set of MDHs from those they use during O$_2$ respiration.

**Methane monooxygenase.** The genomes of *Methylocella tundrae* T4 and *Methylacidiphilum caldifontis* IT6 contain genes that encode sMMO and pMMO, respectively. In the suboxic CH$_4$ + O$_2$ + N$_2$O conditions, all the genes (*mmoXYBZDCRG*) in the gene cluster T4_01946–54 displayed a high degree of transcriptional upregulation (18.7–96-fold) compared to O$_2$-replete CH$_4$ conditions (Supplementary Fig. 9, Supplementary Datasets 5, 6). In a previous study[85], *Methylosinus trichosporium* OB3b sMMO activity and protein expression were found to be significantly elevated under hypoxic conditions (24 μM) compared to higher O$_2$ conditions (188 μM). Furthermore, *Methylosinus trichosporium* OB3b sMMO's catalytic activity in the degradation of dichloroethane was enhanced at low O$_2$ levels and impaired at elevated O$_2$ levels[86]. Thus, in methanotrophs, upregulation of methane monooxygenase genes under O$_2$ limiting conditions might be a strategy to produce more methane monooxygenase. This will lead to increased methane oxidation and thus provide stronger competition for the limited O$_2$ with the terminal oxidase. Aside from the methane monooxygenase genes, group II and III truncated hemoglobin encoding genes were upregulated in *Methylocella tundrae* T4 (T4_02445, T4_02637, and T4_00400; 4- to 12-fold) and *Methylacidiphilum caldifontis* IT6 (IT6_00149; 3-fold) cells in response to suboxia or anoxia (Supplementary Datasets 5, 6). These truncated hemoglobins are thought to transport O$_2$ to the methane monooxygenase[22]. Compared to methane, methanol resulted in lower transcript levels of sMMO genes in *Methylocella tundrae* T4 (Supplementary Fig. 11, Supplementary Dataset 5), with much lower levels in the O$_2$ replete CH$_3$OH + O$_2$ conditions compared to the anoxic CH$_3$OH + N$_2$O conditions (Fig. 6, Supplementary Datasets 5, 6). Transcriptional repression of sMMO genes by growth substrates other than methane has been observed in *Methylocella silvestris* BL2 (refs. 87,88). The expression of

genes encoding denitrification enzymes, their transcriptional regulators, and terminal oxidase is described in Supplementary Note 3.

## Ecological relevance
Our findings revealed that certain methanotrophic strains, particularly those from the genera *Methylocella* and *Methylocystis*, which are commonly found in acidic and neutral terrestrial environments based on ecological meta-data from the BacDive database[89,90], have the ability to reduce N$_2$O. Wetlands, such as acidic peatlands and paddy fields, are significant contributors to the release of CH$_4$ and N$_2$O (refs. 27,91,92). Although active N$_2$O consumption has been observed in acidic wetlands[93], little is known about the microbial mechanisms that drive these processes. In a recent study[27] wherein active N$_2$O consumption was observed in peatlands (pH 6.4–3.7) located in Central and South America, *Methylocystis* species accounted for over 20% of the N$_2$O-reducing microbial community based on *nosZ* gene amplicon sequence variants. This implies that N$_2$OR-containing methanotrophs might make significant contributions to N$_2$O reduction in these environments. The current prevailing perception of N$_2$OR containing methanotrophs as a phylogenetically narrow group with limited ecological impact might be heavily biased by the scarcity of cultured methanotrophs with such metabolic capabilities. Thus, additional in situ and ecogenomic-based investigations are needed to more precisely quantify the contribution of known methanotrophs to N$_2$O reduction as well as to uncover other novel N$_2$O-reducing methanotrophs, such as those belonging to the *Gemmatimonadota* phylum[50].

Short-term or seasonal water table fluctuations caused by either natural or anthropogenic desiccation influence the transition zone from oxic to anoxic conditions in wetlands[94–96]. In the deeper, water-filled anoxic layer of wetlands[97], and even in oxygenated wetland soils[98], methanogens produce CH$_4$. N$_2$O can be produced from denitrification processes, especially by incomplete denitrifiers which are frequently abundant in environments[30–32]. Nitrifiers also produce a significant amount of N$_2$O as a byproduct of ammonia oxidation in the suboxic layers[99]. Furthermore, NO$_2^-$ produced from nitrogen cycling processes can be abiotically reduced to N$_2$O through chemodenitrification due to the stability of Fe$^{2+}$ in acidic peat soils. At the oxic-anoxic interface, where CH$_4$ and O$_2$ gradients overlap, N$_2$O-respiring methanotrophs will have simultaneous access to both CH$_4$ and N$_2$O. Although the CH$_4$-O$_2$ counter gradient is dynamic and O$_2$-respiring organisms can rapidly deplete the limited O$_2$, these N$_2$O-respiring methanotrophs can use a growth strategy that involves respiring both N$_2$O and O$_2$ and coupling it to CH$_4$ oxidation. This unique lifestyle, combined with the potential ability to respire N$_2$O solely with non-methane substrates such as C1, C-C compounds[51,100] as well as H$_2$ (refs. 52,100), can confer a selective growth advantage, facilitate their niche expansion to suboxic and anoxic zones, and make them resilient in such environments.

In conclusion, we revealed that sMMO- and pMMO-containing acidophilic methanotrophs of the genera *Methylocella* and *Methylacidiphilum* can grow anoxically by respiring N$_2$O using clade I and II NosZ, respectively. N$_2$O reduction was detected at an extremely acidic pH of 2.0, which is by far the lowest pH reported for this process[27,92]. Further, N$_2$O reduction can improve the growth yields of these bacteria under O$_2$-limiting conditions and provide a competitive advantage. This study significantly expands our perception of the potential ecological niches of aerobic methanotrophs. In addition to mitigating CH$_4$ and CO$_2$ emissions, aerobic methanotrophs potentially play a role in reducing the emission of the climate-active and ozone-depleting gas N$_2$O, particularly in low pH environments.

## Methods
### Bacterial strains and growth conditions
The methanotrophic bacterial strains used for the experiments include *Methylacidiphilum caldifontis* IT6, *Methylacidiphilum infernorum* IT5,

*Methylocella tundrae* T4 ( = KCTC 52858 [T]), *Methylocella silvestris* BL2 ( = KCTC 52857 [T]), *Methylocystis* sp. SC2, and three in-house *Methylocystis echinoides*-like isolates (strains IM2, IM3, and IM4). The *Methylacidiphilum* strains are also in-house strains isolated previously from a mud-water mixture taken from Pisciarelli hot spring in Pozzuoli, Italy[44]. The *Methylocella* strains were obtained from the Korean Collection for Type Cultures (KCTC). Growth of the bacterial strains was performed using a low salt mineral (LSM) medium. The medium contained 0.4 mM $MgSO_4 \cdot 7H_2O$, 0.2 mM $K_2SO_4$, and 0.1 mM $CaCl_2 \cdot 2H_2O$ and was supplemented with filter-sterilized solutions of 2 mM $(NH_4)_2SO_4$, 0.1 mM $KH_2PO_4$, 1 μM $CeCl_3$, 1 μM $LaCl_3$, 1 mL (1×) vitamin, and 1 mL (1×) trace element solutions[101] per liter. The pH of the medium was adjusted to pH 2.0 with concentrated sulfuric acid (filter-sterilized) for the *Methylacidiphilum* strains and to pH 5.5 with 20 mM 2-morpholinoethanesulfonic acid (filter-sterilized) for the *Methylocella* strains. The cultures were incubated at 52 °C for *Methylacidiphilum* strains (IT5 and IT6) and 28 °C for *Methylocella* strains (T4 and BL2) with shaking at 160 rpm. Unless stated otherwise, ammonium sulfate, $(NH_4)_2SO_4$, was used as the nitrogen source.

## Enrichment and isolation of *Methylocystis* strains

The $N_2OR$-containing *Methylocystis* strains were isolated from an acidic forest soil in Chungcheongbuk-do, South Korea (36°55'31" N 127°54'86" E). The soil sample preparation and initial enrichment of the methanotrophs[102], as well as the isolation of methanotrophic strains through repeated serial dilution of the enrichment cultures[103], have all been described previously. Briefly, the most diluted culture exhibiting methane oxidation was serially diluted and filtered through 0.2-μm Track-Etch membrane polycarbonate filters (Whatman). The filters were placed on LSM medium (pH 5.5) in Petri dishes and incubated at 30 °C in airtight containers containing $CH_4$ (10%, v/v) and $CO_2$ (5%, v/v). Colonies that appeared on the filters after 3 weeks of incubation were transferred to fresh LSM medium in 160-mL serum vials with the same gas composition. Three individual methanotrophic isolates were identified by sequencing the 16 S rRNA gene with the 27 F/1492 R primer set[104]. The purity of the isolates was confirmed by seeding aliquots of the $CH_4$-grown cultures into the LSM medium with 0.05% (w/v) yeast extract, tryptic soy broth, and Luria-Bertani broth without $CH_4$ and incubating at 30 °C. Three methanotrophic isolates, IM2, IM3, and IM4, shared 99.46% 16 S ribosomal RNA (rRNA) gene-sequence identity with the alphaproteobacterial methanotroph *Methylocystis echinoides* LMG27198. The three strains share average nucleotide identity values ranging from 81.85–81.93 with *Methylocystis echinoides* LMG27198, implying that they represent a new species in the genus *Methylocystis*.

## DNA isolation, genomic and phylogenetic analyses

High-molecular-weight genomic DNA was extracted using a modified CTAB method[105], from 200 mL amounts of *Methylocella tundrae* T4 grown in methanol, and the *Methylocystis* isolates (strain s IM2, IM3, and IM4) grown in $CH_4$. The genomes of *Methylocella tundrae* T4 and *Methylocystis* sp. IM3 were sequenced at LabGenomics (Seongnam, Republic of Korea) and Macrogen (Seoul, Republic of Korea) using the PacBio RS II (long-read sequencing) and Illumina HiSeq (2 × 150 bp) platforms, respectively. The genomes of *Methylocystis* sp. IM2 and *Methylocystis* sp. IM4 were sequenced using a MinION R10.4.1 flow cell (FLO-MIN114, Oxford Nanopore Technologies). The PacBio reads were assembled with the Trycycler pipeline (v0.5.4)[106]. Filtered reads were subsampled and assembled using Miniasm/Minpolish (v0.3-r179)[107], Flye (v2.9.2)[108], and Raven (v1.8.3)[109] assemblers. The consensus contigs were polished with Illumina short reads using Polypolish (v0.5.0)[110] and POLCA (v4.0.5)[111]. The circularity was confirmed during the Trycycler pipeline assembly and again by mapping the Illumina reads backward. De novo genome assembly of the MinION long reads was accomplished using Flye (v2.9.2)[108]. Annotation of methanotrophs' genomes was performed with the Prokka annotation pipeline

(v1.14.6)[112] and NCBI Prokaryotic Genome Annotation Pipeline (PGAP; v4.2)[113]. Functional assignment of the predicted genes was improved using a set of public databases (InterPro[114], GO[115,116], PFAM[117], CDD[118], TIGRFAM[119], and EggNOG[120]). Prediction of signal peptides and trans-membrane helices was performed using the web-based services SignalP (v6.0)[121] and TMHMM (v2.0)[122] with default settings.

The distribution of denitrification genes in methanotroph isolates or metagenome-assembled genomes (MAG) (meeting the following CheckM (v1.2.2) criteria: completeness > 60% and contamination <10%) was examined using genomic data from the NCBI assembly database. Reference protein sequences of denitrification enzymes (NapA, NapB, NarG, NarH, NarI, NirS, NirK, NorB, NorC, and NosZ) were obtained from the NCyc[123] and BV-BRC[124] databases. The annotated protein sequences of methanotrophs were re-annotated against the obtained reference sequences from the NCyc[123] and BV-BRC[124] databases. The identities of the obtained denitrification protein sequences in methanotrophs were verified using manual alignment and tree-building procedures with reference sequences. Sequences incorrectly annotated as denitrification genes were removed, and only candidate genes that clustered with reference sequences were counted as true hits.

For phylogenetic analyses of the NosZ proteins and methanol dehydrogenases of strains T4 and IT6, representative amino acid sequences of the genes of related taxa were obtained from NCBI. The derived amino acid sequences of the NosZ and methanol dehydrogenases (XoxF and MxaF) were aligned using MAFFT (v7.511)[125]. Maximum-likelihood trees were inferred with IQ-TREE (v1.6.12). The constructed trees and operon arrangements were visualized using iTOL (v.6.7.2)[126] and used for annotation. Genomic islands were predicted using the IslandViewer web server[127].

## Anoxic growth coupled with $N_2O$ reduction

To demonstrate the ability of $N_2OR$-containing methanotrophs to grow using $N_2O$ as the electron acceptor, we established anoxic batch cultures of *Methylocella tundrae* T4 and *Methylacidiphilum caldifontis* IT6 in 160-mL bottles containing 20 mL of LSM media and inoculated with 1–5% (v/v) actively growing-cells from the log phase (starting $OD_{600}$ values ≤ 0.05). To remove oxygen, nitrogen gas ($N_2$, purity >99.999%) was introduced into the bottles via a long needle (18 G). Following that, the bottles were flushed with $N_2$ gas for 20 min before being sealed with gas-tight butyl rubber stoppers and aluminum crimp seals to prevent $O_2$ leakage. We used contactless trace-range oxygen sensor spots (TROXSP5) to monitor $O_2$ contamination (<0.10%, v/v) in the culture bottles incubated after $N_2$-flushing (see *Analytical methods*, for details). These spots have a detection limit of 20 nM $O_2$. Chemical-reducing agents, $Na_2S$ (0.5, 1, and 2 mM), cysteine (0.5 mM), DTT (0.5 mM), and titanium citrate (0.5 and 1 mM) in the media resulted in severe cell toxicity, hindering their use in this study as previously reported for $N_2OR$ reducer *Anaeromyxobacter dehalogenans*[128]. When the cultures were incubated without the chemical-reducing agents, the cells completely depleted the trace $O_2$ concentration present in the culture bottles in less than 24 h as measured by the oxygen sensor spots.

The $N_2OR$-lacking methanotrophs *Methylocella silvestris* BL2 and *Methylacidiphilum infernorum* IT5, which are closely related to *Methylocella tundrae* T4 and *Methylacidiphilum caldifontis* IT6, respectively, were used as negative controls. Methanol (30 mM), $N_2O$ (5%, v/v), and $CO_2$ (5%, v/v) were used as the energy source, electron acceptor, and carbon source, respectively. In addition, pyruvate (10 mM) and hydroxyacetone (acetol) (10 mM) were tested as the sole C-C electron donors in strains T4 and IT6, respectively. Furthermore, strain IT6 cells were investigated to grow chemolithoautotrophically in sealed 1-liter bottles (duplicate) containing 20 mL of LSM medium at pH 2.0 on $H_2$ (10% v/v) with and without $N_2O$ (5% v/v). As part of the control experiments, we incubated cells from the four strains in LSM

media under anoxic conditions (without N$_2$O) to assess the contribution of the initial trace O$_2$ present in the culture bottles to biomass increase. The increase in biomass as OD$_{600}$ by the trace O$_2$ in the control cultures was negligible when compared to cultures growing with N$_2$O as the sole electron acceptor (see Fig. 2C, F, I, L). Positive control experiments with methanol (30 mM) and O$_2$ (5%, v/v) as the electron donor and electron acceptor, respectively, were conducted for each strain. The concentrations of H$_2$, O$_2$, N$_2$O, NO$_3^-$, and NO$_2^-$ were monitored at intervals during incubations (described in *Analytical methods*). Cell growth was also evaluated using optical density measurements (λ = 600 nm), direct microscopic cell counts, and real-time quantification of 16 S rRNA gene abundance (described in *Analytical methods*). All growth experiments were performed in triplicates unless otherwise stated.

Next, we checked the anoxic growth of *Methylocella* strains on NO$_3^-$ (2 to 4 mM KNO$_3$) as the terminal electron acceptor instead of N$_2$O. Methanol (30 mM) was used as the sole electron donor and 2 mM NH$_4^+$ was used as the N-source. To compare the effect of electron donors on NO$_3^-$ and NO$_2^-$ reduction, *Methylocella* strains were also anoxically grown in LSM medium containing a C-C substrate, pyruvate (10 mM). Cells of strain T4 were grown under O$_2$-replete (O$_2$; 21%, v/v) or anoxic conditions (O$_2$; 0%, v/v, N$_2$O; 5%, v/v) for the NO$_2^-$ toxicity test (triplicates) with varying NO$_2^-$ (KNO$_2$) concentrations (0, 0.01, 0.03, 0.1, 0.3, and 1 mM).

## Analytical methods

A YL 6100 gas chromatograph (YL Instrument Co., Anyang, South Korea) with a flame ionization detector (FID) and a thermal conductivity detector (TCD) was used to analyze the mixing ratios of CH$_4$, N$_2$O, and H$_2$ in the headspace of the sealed bottles used to cultivate the *Methylocella* and *Methylacidiphilum* strains. Using a Hamilton glass syringe, 100 µL of the sealed bottle headspaces were injected into a gas chromatograph equipped with MolSieve 5 A column (3Ft, 1/8, 2 mm, 60/80 SST, Agilent Technologies, Inc., CA, USA; for separating H$_2$, O$_2$, and N$_2$O) and Haysep N column (7Ft, 1/8, 2 mm, 60/80 SST, Agilent Technologies, Inc., CA, USA; for separating CO$_2$ and CH$_4$) to determine the gases present. Helium was used as the carrier gas, with a flow rate of 15 mL·min$^{-1}$. By utilizing pure gases of known concentrations, a calibration curve of the gases used as substrates was generated. The bottles were fitted with contactless trace range oxygen sensor spots (TROXSP5, PyroScience, Germany) calibrated at 0% and ambient air (21% oxygen), and a FireSting-Pro multi-analyte meter (FSPRO-4, PyroScience, Germany) was used to measure the O$_2$ concentration in the sealed bottles. Acidic Griess reagent and VCl$_2$/Griess reagent were used for photometric quantification of NO$_2^-$ and NO$_3^-$ concentrations[129], respectively, using a SpectraMax M2 microplate reader (Molecular Devices, USA). Cell growth was assessed by measuring changes in OD$_{600}$ using a spectrophotometer (Optizen 2120UV, Mecasys Co., Daejeon, Korea). Real-time quantification of the 16 S rRNA gene was performed using the 518 F/786 R primer set[130]. The total cell number was determined by counting cells stained with DAPI (4,6-diamidino-2-phenylindole) using an epifluorescence microscope (AxioScope.A1; Carl Zeiss, Oberkochen, Germany).

## Kinetic analysis using microrespirometry (MR)

For kinetic analysis using microrespirometry (MR), *Methylocella tundrae* T4 cells were grown under three different O$_2$ conditions: O$_2$-replete (CH$_3$OH + O$_2$), suboxic (CH$_4$ + O$_2$ + N$_2$O), and anoxic (CH$_3$OH + N$_2$O). *Methylacidiphilum caldifontis* IT6 cells were grown under O$_2$-replete (CH$_3$OH + O$_2$) and anoxic (CH$_3$OH + N$_2$O) conditions. The O$_2$-replete growth conditions included ambient air (21% O$_2$, v/v) and CH$_3$OH (30 mM) as the sole electron donor. The suboxic cell cultures were grown under a condition that included CH$_4$ (5%, v/v) as the sole electron donor and O$_2$ (0.5%, v/v) with N$_2$O (1%, v/v) as terminal electron acceptors. O$_2$ (0.5%, v/v) was resupplied intermittently before

its depletion. Anaerobically grown cells were cultured in bottles containing 30 mM CH$_3$OH as the sole electron donor and 5% (v/v) N$_2$O as the terminal electron acceptor. The cultures were monitored daily and harvested as soon as active consumption of electron donors and acceptors was detected. After being collected by centrifugation (5000 × g, 30 min, 25 °C), the cells were washed twice with substrate- and N-source-free MES-buffered LSM (20 mM MES; pH 5.5) or H$_2$SO$_4$-buffered LSM (4 mM H$_2$SO$_4$; pH 2.0) and then resuspended in 20 mL of the same media without electron donors and acceptors. In the cultures grown under anoxic and suboxic conditions, the cell suspensions were transferred to sealed 20-mL bottles and flushed with nitrogen gas (N$_2$, purity >99.999%) before use. The cell suspensions were dispensed into a double-port MR chamber (no headspace) with a capacity of 5 or 10 mL outfitted with O$_2$ and N$_2$O-detecting microsensors, two MR injection lids, and two glass-coated stir bars. Kinetics and stoichiometry of N$_2$O and O$_2$ reduction coupled to CH$_3$OH oxidation were estimated using anoxic CH$_3$OH + N$_2$O- and oxic CH$_3$OH + O$_2$-grown cells, respectively. Anoxic CH$_3$OH + N$_2$O-grown cells were used to test CH$_3$OH-dependent O$_2$ and N$_2$O uptake by strains IT6 (starting OD$_{600}$ = 0.96) and T4 (starting OD$_{600}$ = 0.79). The effect of O$_2$ to N$_2$OR activities of strains T4 and IT6 was determined by spiking varying O$_2$ to the N$_2$O respiring cells. In a 5-mL MR chamber, suboxic CH$_4$ + O$_2$ + N$_2$O-grown cells of strain T4 (starting OD$_{600}$ = 1.0) were used to test the CH$_4$-dependent simultaneous respiration of O$_2$ and N$_2$O.

All MR experiments were performed in a recirculating water bath at 27 °C and 50 °C for strains T4 and IT6, respectively. A 10-µL or 50-µL syringe (Hamilton, Reno, USA) fitted with a 26 G needle was used to inject the substrate (CH$_4$, CH$_3$OH, N$_2$O, or O$_2$) into the chamber via an injection port. Concentrations of O$_2$ and N$_2$O were measured using an OX-MR oxygen microsensor (OX-MR-202142, Unisense, Aarhus, Denmark) and a N$_2$O-MR sensor (N2O-MR-303088, Unisense), respectively. The detection limits of the OX-MR and N$_2$O-MR microsensors are 0.3 µM O$_2$ and 0.1 µM N$_2$O, respectively. The OX-MR and N$_2$O-MR microsensors were directly plugged into a microsensor multimeter before being polarized for more than a day and calibrated according to the manufacturer's instructions. All data from the microsensor multimeter was logged onto a laptop using SensorTrace Suite software (v.3.3.0, Unisense). Anoxically prepared aliquots of N$_2$O, CH$_4$, and CH$_3$OH were injected into the MR chamber via the injection port with a 10-µL syringe (Hamilton, Reno, USA). Anoxic substrate-free LSM media (at pH 2.0 and 5.5) were prepared by sparging the solutions with N$_2$ gas for 1 h before use. Anoxic saturated-aqueous CH$_4$ and N$_2$O solutions were made in capped 160-mL bottles containing 100 mL of LSM medium and pressurized with CH$_4$ or N$_2$O (1, 2, or 3 atm; 100%, v/v). Saturated-aqueous O$_2$ solutions were prepared in capped 160-mL bottles containing 100 mL of LSM medium and pressurized with O$_2$ (1, 2, and 3 atm; 100%, v/v).

## Growth based on CH$_4$ oxidation coupled with co-respiration of O$_2$ and N$_2$O

Suboxic cultivations were carried out to investigate the growth of *Methylocella tundrae* T4 by oxidizing methane with simultaneous respiration of O$_2$ and N$_2$O. The experiments were conducted in N$_2$-flushed 2-liter sealed bottles containing 60 mL of LSM medium with 2 mM NH$_4^+$ as the N-source. The headspace of the bottles was composed of CH$_4$ (5%, v/v), O$_2$ (0.5%, v/v), N$_2$O (1%, v/v), and CO$_2$ (5%, v/v) and supplemented with additional O$_2$ (-0.5%, v/v) before its depletion. The headspace gas (CH$_4$, N$_2$O, and O$_2$) mixing ratios were monitored at intervals during incubations as described above in *Analytical methods*. To investigate the growth benefits of cells of strain T4 respiring N$_2$O in tandem with O$_2$ during CH$_4$ oxidation, an O$_2$-replete culture was included for comparison (triplicates). The apparent increase in cell densities of both growth conditions was compared using OD$_{600}$ measurements.

## Transcriptome analysis

Cells of strains T4 and IT6 were cultured in 60 mL of LSM medium at pH 5.5 and pH 2.0 in sealed 2-liter bottles (4 or 5 replicates) for transcriptome analyses. Strain T4 cells were cultured under three different $O_2$ levels, with the first setting being $O_2$-replete ($CH_4 + O_2$ and $CH_3OH + O_2$), the second being suboxic ($CH_4 + O_2 + N_2O$), and the third being anoxic ($CH_3OH + N_2O$). Strain IT6 was cultivated in $O_2$-replete $CH_3OH + O_2$ and anoxic $CH_3OH + N_2O$ conditions. Cells were grown anaerobically in bottles containing 30 mM $CH_3OH$ as the sole electron donor and 5% $N_2O$ as the terminal electron acceptor. The $O_2$-replete growth conditions were made up of ambient air (21% $O_2$, v/v) with $CH_4$ (5%, v/v) or $CH_3OH$ (30 mM) serving as the sole electron donor. The suboxic growth conditions were made up of a mixture of $CH_4$ (5% v/v) as the sole electron donor and $O_2$ (0.5% v/v) and $N_2O$ (1% v/v) as terminal electron acceptors. Before the depletion of $O_2$, additional $O_2$ was resupplied intermittently at a mixing ratio of 0.5% (v/v). Contactless trace-range oxygen sensor spots (TROXSP5) were installed into the culture bottles to monitor $O_2$ concentration.

The cells were harvested during the mid-exponential phase by centrifugation at $5000 \times g$ for 10 min at 25 °C. Total RNA was extracted from the cells in four replicates using the AllPrep DNA/RNA Mini Kit (Qiagen) according to the manufacturer's protocol. RNA quality was checked with the Agilent 2100 Expert Bioanalyzer (Agilent), and cDNA libraries were prepared from the RNA samples using the Nugen Universal Prokaryotic RNA-Seq Library Preparation Kit. The cDNA libraries were sequenced using NovaSeq6000 (Illumina) at LabGenomics (Seongnam, Korea). Read quality was evaluated with FastQC (v0.11.8)[131]. Trimmomatic (v0.36)[132] was used to trim reads with the options: SLIDINGWINDOW:4:15 LEADING:3 TRAILING:3 MINLEN:38 HEADCROP:13. Reads mapped to strains T4 and IT6 rRNA sequences were removed with SortMeRNA (v4.3.6)[133]. The remaining reads were aligned to the genomes of strains T4 and IT6 using Bowtie2 (v2.4.4)[134], and the reads mapped to each gene were counted using HTSeq (v0.12.3)[135]. Expression values are presented as transcripts per kilobase million (TPM). The statistical analysis of differentially expressed genes was performed using the DESeq2 package in R (v4.3.2). A two-sided Wald test was used to calculate the $p$ values, and multiple-comparison adjustments were made using the Benjamini-Hochberg method by default in DESeq2 (v1.40.2).

## Reporting summary

Further information on research design is available in the Nature Portfolio Reporting Summary linked to this article.

# Data availability

All numerical data used to make the figures is provided in source data. The complete genome sequence of strain T4 was deposited in the National Center for Biotechnology Information (NCBI) GenBank (accession nos. CP139089 (Chromosome), CP139088 (Plasmid 1), and CP139087 (Plasmid 2)). The genomic sequences and genome annotations of *Methylocystis* species (strains IM2, IM3, and IM4) and 'Ca. Methylotropicum kingii' are available on Figshare (https://doi.org/10.6084/m9.figshare.25521913.v2). All previously sequenced genomes analyzed in this study are available in the NCBI Database with the GenBank accession numbers listed in Supplementary Dataset 1. The whole transcriptome data was deposited in the NCBI BioProject database under the accession number PRJNA1050235. The following are the publicly available databases/datasets used in the study: NCBI NR [https://www.ncbi.nlm.nih.gov/refseq/], BV-BRC, NCyc [https://github.com/qichao1984/NCyc], Pfam [https://pfam.xfam.org/], InterPro [https://www.ebi.ac.uk/interpro/], GO [https://geneontology.org/], CDD, TIGRFAM, and EggNOG [https://tigrfams.jcvi.org/cgi-bin/index.cgi]. Source data are provided with this paper.

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

## Acknowledgements

This work was supported by the NRF (National Research Foundation of Korea) grant funded by the Korean government (Ministry of Science and ICT) (2021R1A2C3004015), the Basic Science Research Program through NRF funded by the Ministry of Education (2020R1A6A1A06046235), and

the National Institute of Agricultural Science, Ministry of Rural Development Administration, Republic of Korea (research project PJ01700703). J.-H.G. was supported by the NRF grant funded by the Korean government (Ministry of Science and ICT) (RS-2023-00213601). M.-Y.J. was supported by the NRF grant funded by the Korean government (Ministry of Science and ICT) (2021R1C1C1008303 and 2022R1A4A503144711). P.F.D. was supported by a Natural Sciences and Engineering Research Council of Canada (NSERC) Discovery Grant (2019-06265). M.W. was supported by the Austrian Science Fund FWF Cluster of Excellence "Microbiomes drive planetary health" COE7.

## Author contributions

S.I.A., J.-H.G., and S.-K.R. designed research. S.I.A., J.-H.G., M.-Y.J., and Y.K. performed research. S.I.A., J.-H.G., Y.K., M.-Y.J., P.F.D., and S.-K.R. analyzed data. S.I.A., J.-H.G., M.-Y.J., P.F.D., M.W., and S.-K.R. wrote the manuscript with contributions and comments from all co-authors.

## Competing interests

The authors declare no competing interests.
