## [Peer Review File · Nature Communications]

Nitrous oxide respiration in acidophilic methanotrophsReviewer #1 (Remarks to the Author):

Noteworthy results: This manuscript examined the capability of three aerobic methanotrophs in the *Methylocella* and *Methyloacidiphilum* genera to respire N-oxides under low or no oxygen. *Methylocella tundrae* T4 and *Methylocella silvestris* BL2 were capable of growing in the absence of oxygen on nitrate/nitrite as electron acceptors with pyruvate as an electron donor. *Methyloacidiphilum caldifontis* IT6 is an extremophile and was able to respire N₂O and oxidize methane under hypoxic conditions at high temperature and pH 2.0. *Methylocella tundrae* T4 could respire N₂O and oxidize methane under hypoxic conditions, and grow on methanol and N₂O under anoxic conditions, at neutral pH. Gene expression was linked to these metabolic shifts. The ability of the methanotroph strains to connect carbon oxidation to nitrogen oxide reduction expands the repertoire of known methanotroph physiology and shows their capacity to consume nitrous oxide as well as methane under certain conditions. This is a novel finding showing that particular strains of methanotrophs can grow from N₂O respiration, but not with methane as a carbon source (only methanol and pyruvate).

Significance to field: Previous studies have connected methane and methanol oxidation to the reduction of nitrate/nitrite in aerobic methanotrophs. This study is an expansion showing growth of specific strains of methanotrophs using N-oxides as electron acceptors, and also the capacity to connect single carbon oxidation to nitrous oxide reduction under hypoxic (methane) to anoxic (methanol) conditions. The strains identified to possess NosZ are fairly restricted phylogenetically and may be somewhat limited to a relatively small niche space. Thus, their ability to impact GHG emissions is questionable, but the discovered pathways are novel and impactful for understanding physiological diversity in this group of microorganisms and showing that they can grow using N₂O as an electron acceptor. Furthermore, the cultivation work demonstrated the reduction of N₂O at pH 5.5 and 2.0, which is an exciting observation to follow up on as N₂O reduction is generally suppressed in acidic soils.

Support of conclusions/claims: The phylogenetic analysis of NosZ genes across methanotrophs is extensive, well supported, and clearly presented. The cultivation studies are well controlled and the data are validated with multiple line of evidence. Aside from carbon substrates, another experiment showed that *M. caldifontis* IT6 could also grow on hydrogen and N₂O, which is a "bonus" physiological discovery. This metabolism was validated by linking growth with the expression of a NiFe hydrogenase. In the conclusion statements, it would be helpful to put this physiology into an ecological context as the microorganisms that have the ability seem rather taxonomically and ecologically limited. Although they have capacity to reduce N₂O to N₂, how/when would this physiology be significant in nature and how much could they add to the N₂O sink?

Data analysis, interpretation and conclusion: The data analysis is comprehensive and covers gaps in the genomic presence and organization of NosZ gene clusters, reasons why nitrate/nitrite reduction does not support growth of methanotrophs under hypoxia to anoxia (RNS accumulation), and a demonstration of robust N₂O reduction linked to cognate gene expression and growth with methanol as a substrate (Fig. 6 metabolic map).

The authors should consider changing the header on line 339 to "N₂O reduction coupled with CH₃OH or CH₄ oxidation as the comparison between growth on both C1 substrates is described, but methanol is the main substrate supporting N₂O reduction under anoxic conditions.

Methodologies: the methods used are comprehensive and up-to-date for the field. There is ample information included in each subsection to clearly understand the experimental set-up and to replicate the experiments. There were enough replicated physiological and transcriptomic experiments to validate the observations.

A modeling exercise at the end would be helpful to put into context the ecological impact of this metabolism given the taxonomic and niche limitations of the NosZ-encoding groups, and also the conditions that amplify methane oxidation versus those that are conducive to N₂O reduction. These two activities seem to be separated in these microbes as governed by substrate availability, redox potential, and regulatory factors.

Reviewer #2 (Remarks to the Author):

In addition to carbon dioxide, methane and nitrous oxide are the most important greenhouse gases contributing to global warming. Although theoretically there should be microbes that can couple methane oxidation to nitrous oxides reduction independently, as have been suggested by genomic surveys, experimental demonstration of this feasibility have not been reported. In this study, the authors show that some acidophilic methanotrophs can grow anaerobically on N₂O and methanol, and can oxidize CH₄ and reduce N₂O under hypoxic conditions. It is a significant progress in our understanding of the metabolic potentials of these methanotrophs, especially in the acidic environments.

The experiments are well-designed and conducted, with all necessary controls to support the conclusions. The manuscript is well organized and written. I have several comments for the authors to address during revision:

Some of the transcriptomic results are confusing, e.g., expressions of the genes responsible for methane oxidation and nitrate reduction were upregulated, when these conversions were not there. Although the authors have tried to give some potential explanation by reviewing literatures reporting similar or related observations, could a better approach be expanding the transcriptomic analysis to samples from other batch tests where methane or NO_x were and were not supplied? Basically more control groups for contrast, and more chance to eliminate potential batch effect.

In order to figure out why there was no growth of these methanotrophs when respiring on nitrate or nitrite, the authors carried out several experiments to link it to the accumulation of nitrite, and potential accumulation of C1 metabolites. I am wondering if the accumulation of NO could be another inhibitory factor. Checking the NO profile during the tests and even spike NO at different concentrations will not only clear up this point, but also shed light on the NO respiration of methanotrophs.

Minor comment - there are some issues with references in SI, e.g., line 37, 189, 207.

Reviewer #3 (Remarks to the Author):

This manuscript is very interesting and does give novel insights into the degree to which metabolic boundaries as we know them, can be stretched. I do think the notion that growth can be supported from N₂O as electron acceptor is very interesting.

However, for the purposes of applications, I would encourage the authors to expand the sections where an application is promised or suggested. In particular where, can this be applied under the conditions methanotrophs work: methane rich conditions, and where only nitrous oxide is available to avoid nitrite toxicity? The experiments were heavily manipulated with customized conditions and electron donors that are not really what methanotrophs have in nature.

Some specific comments:

Line 47: Authors should remove the application promise, even the core metabolisms of nitrate or nitrite dependent methane oxidation, have been promised since 2011 and still only a handful of pilot attempts with average performance have been achieved. Applications should not be oversold, and remain as ecological significance of the metabolism.

Line 55: the super index for references is not consistent, sometimes they have more spacing than others. Line 65.

Line 57: methane is not estimated at 34 times, authors should update references.

Line 99: punctuation is odd, there needs to be a revision for ":"

Line 211: the term higher growth rates with higher values, means they grow slower. This nuance

should be better evidenced in clearer writing.

Line 236: the authors should include more details on the growth, these strains under the media used, are reported for the yield using DW measurements. What is this dry weight? Just bacterial cells? Is there any EPS produced? Is there any mass interference that can over estimate the yield? Methanotrophs are hardly pure and planktonic in all environments, and I am concerned with the yield relying on weight. Cell numbers with counting, or with housekeeping 16S copy increase could have been better. In addition, FISH visual representation of the increase of specific % of few observations could evidence directly the yield truly comes from an increase in actual cells. Same applies for the growth, is based on OD values, and this is just an optical detection of light through a sample, but are cell numbers really increasing to the rates detected? Could the cells be storing and releasing carbon biopolymers, was the culture "flocculent" after some time? Was it really free-living planktonic cell units? I would appreciate, some insights into this.

260: what is not clear to me at this point is, why base all on methanol? Methanol oxidation requires more energy in the cell as when starting with the intended goal of methane activation. I get that your experiments are not based on oxygen dependent activation, and that is why the conditions are altered from methanol to nitrous oxide as limits of the system. As valuable as it is that under these conditions, the growth from nitrous oxide as acceptor happens, the applicability or real occurrence may not be as significant as the results in vitro suggests.

272: The growth dependent on the carbon fixing pathways that was observed under nitrous oxide, could then continue following the depletion of nitrate or nitrite reduction. This seems to not have been pursued, even under the same methanol conditions as energy source.

Line 304: I would like to see more relation to real ecology, where will an environment where nitrite does not come from nitrate reduction and only nitrous oxide is the only available electron acceptor can be found?

613: if authors want to speculate on applications, this could be expanded as to how, under what conditions and provide insights so the field can follow this line of thought. Else, I would encourage the authors form suggesting applications in engineered systems.

Work is reproducible, methodology is sound, and only the yield and growth estimations have some observations from my part for clarity.

I end by stating that the experiments are diverse, complete, very interesting and overall, the work is relevant for a metabolic perspective, and for fundamental value.

Response to Reviewers' Comments

We express our gratitude to the reviewers for thoroughly reviewing the manuscript and for providing valuable feedback and positive comments. Their input has significantly improved the revised paper we are submitting. We have included our responses to the reviewers' comments as inserts (blue text) below:

Reviewer #1 (Remarks to the Author):

- **Noteworthy results:** This manuscript examined the capability of three aerobic methanotrophs in the *Methylocella* and *Methyloacidiphilum* genera to respire N-oxides under low or no oxygen. *Methylocella tundrae* T4 and *Methylocella silvestris* BL2 were capable of growing in the absence of oxygen on nitrate/nitrite as electron acceptors with pyruvate as an electron donor. *Methylacidiphilum caldifontis* IT6 is an extremophile and was able to respire N₂O and oxidize methane under hypoxic conditions at high temperature and pH 2.0. *Methylocella tundrae* T4 could respire N₂O and oxidize methane under hypoxic conditions, and grow on methanol and N₂O under anoxic conditions, at neutral pH. Gene expression was linked to these metabolic shifts. The ability of the methanotroph strains to connect carbon oxidation to nitrogen oxide reduction expands the repertoire of known methanotroph physiology and shows their capacity to consume nitrous oxide as well as methane under certain conditions. This is a novel finding showing that particular strains of methanotrophs can grow from N₂O respiration, but not with methane as a carbon source (only methanol and pyruvate).
- **Significance to field:** Previous studies have connected methane and methanol oxidation to the reduction of nitrate/nitrite in aerobic methanotrophs. This study is an expansion showing growth of specific strains of methanotrophs using N-oxides as electron acceptors, and also the capacity to connect single carbon oxidation to nitrous oxide reduction under hypoxic (methane) to anoxic (methanol) conditions. The strains identified to possess NosZ are fairly restricted phylogenetically and may be somewhat limited to a relatively small niche space. Thus, their ability to impact GHG emissions is questionable, but the discovered pathways are novel and impactful for understanding physiological diversity in this group of microorganisms and showing that they can grow using N₂O as an electron acceptor. Furthermore, the cultivation work demonstrated the reduction of N₂O at pH 5.5 and 2.0, which is an exciting observation to follow up on as N₂O reduction is generally suppressed in acidic soils.
- **Support of conclusions/claims:** The phylogenetic analysis of NosZ genes across methanotrophs is extensive, well supported, and clearly presented. The cultivation studies are well controlled and the data are validated with multiple line of evidence. Aside from carbon substrates, another experiment showed that *M. caldifontis* IT6 could also grow on hydrogen and N₂O, which is a "bonus" physiological discovery. This metabolism was validated by linking growth with the expression of a NiFe hydrogenase. In the conclusion statements, it would be helpful to put this physiology into an ecological context as the microorganisms that have the ability seem rather taxonomically and ecologically limited. Although they have capacity to reduce N₂O to N₂, how/when would this physiology be significant in nature and how much could they add to the N₂O sink?

RE.) First of all, we express our gratitude for the favorable remarks about our study by the reviewer. Considering the apparent restriction of these methanotrophs to a few known phylogenetic lineages, the reviewer suggested that we discuss their ecological impact in various environments. Following this suggestion, we added information about the ecological significance of these methanotrophs to a new section of the revised manuscript (see Line 528-543). The ecological impact of oxidizing H₂ and other C-C compounds

coupled with N₂O reduction, was also discussed (lines 554-557). We agree that N₂OR-containing methanotrophs are phylogenetically restricted. However, alphaproteobacterial methanotrophs such as *Methylocystis* are frequently ecologically abundant in environments, and we identified them as the primary N₂OR-containing methanotrophs (Supplementary Table 1) and published data indicate that they can make up to 20% of the N₂O-reducing microbial community in some ecosystems¹ (lines 534-538). Further, we believe this phylogenetic restriction is due to the scarcity of cultured methanotrophs possessing this metabolic capability; therefore, this conclusion may be biased. For example, most members of the phylum *Gemmatimonadota* are N₂OR-containing², and a yet-uncultivated N₂OR-containing methanotroph, 'Ca. *Methylotropicum kingii*', also with hydrogenotrophic and C-C utilizing potentials, was postulated recently within this phylum³. This shows that uncultivated facultative methanotrophs with N₂O-reducing capabilities exist. We now discuss these important points in the revised manuscript (lines 538-543).

Data analysis, interpretation and conclusion: The data analysis is comprehensive and covers gaps in the genomic presence and organization of NosZ gene clusters, reasons why nitrate/nitrite reduction does not support growth of methanotrophs under hypoxia to anoxia (RNS accumulation), and a demonstration of robust N₂O reduction linked to cognate gene expression and growth with methanol as a substrate (Fig. 6 metabolic map).

The authors should consider changing the header on line 339 to "N₂O reduction coupled with CH₃OH or CH₄ oxidation as the comparison between growth on both C1 substrates is described, but methanol is the main substrate supporting N₂O reduction under anoxic conditions.

RE.) Thanks for the suggestion. We have changed the header as suggested. Lines 335

- **Methodologies:** the methods used are comprehensive and up-to-date for the field. There is ample information included in each subsection to clearly understand the experimental set-up and to replicate the experiments. There were enough replicated physiological and transcriptomic experiments to validate the observations.

A modeling exercise at the end would be helpful to put into context the ecological impact of this metabolism given the taxonomic and niche limitations of the NosZ-encoding groups, and also the conditions that amplify methane oxidation versus those that are conducive to N₂O reduction. These two activities seem to be separated in these microbes as governed by substrate availability, redox potential, and regulatory factors.

RE.) Thanks for the suggestion. We have revised the manuscript to include an additional section titled "*Ecological relevance*" (lines 528-557) that elaborates on the ecological ramifications, niche advantages, and competitive benefits of N₂O-reducing methanotrophs. We could not include a modeling exercise in the current study; nevertheless, we intend to include it in future studies to better understand under which conditions N₂OR-containing methanotrophs would be most competitive.

Reviewer #2 (Remarks to the Author):

- In addition to carbon dioxide, methane and nitrous oxide are the most important greenhouse gases contributing to global warming. Although theoretically there should be microbes that can couple methane oxidation to nitrous oxides reduction independently, as have been suggested by genomic surveys, experimental demonstration of this feasibility have not been reported. In this study, the authors show that some acidophilic methanotrophs can grow anaerobically on N₂O and methanol, and can oxidize CH₄ and reduce N₂O under hypoxic conditions. It is a significant progress in our understanding of the metabolic potentials of these methanotrophs, especially in the acidic environments.

The experiments are well-designed and conducted, with all necessary controls to support the conclusions. The manuscript is well organized and written. I have several comments for the authors to address during revision:

RE.) We appreciate the reviewer's positive evaluation of our work and value the insightful and constructive feedback provided.

- Some of the transcriptomic results are confusing, e.g., expressions of the genes responsible for methane oxidation and nitrate reduction were upregulated, when these conversions were not there. Although the authors have tried to give some potential explanation by reviewing literatures reporting similar or related observations, could a better approach be expanding the transcriptomic analysis to samples from other batch tests where methane or NO_x were and were not supplied? Basically more control groups for contrast, and more chance to eliminate potential batch effect.

RE.) We thank the reviewer for pointing out the unclear sections in the gene expression data. We reorganized this section in the revised manuscript to avoid such confusion (Lines 508-521). First, we compared the upregulation of sMMO encoding genes in strain T4 cells cultivated under suboxic CH₄ + O₂ + N₂O conditions to O₂-replete CH₄ conditions. We then discussed the physiological implications of this gene upregulation. Second, we compared sMMO gene expression in *M. tundrae* cells grown with methane and methanol. Lower transcript levels of the sMMO genes were observed in the methanol-growth conditions (Supplementary Fig. 11). This is consistent with previous reports on the transcriptional repression of the sMMO genes in *Methylocella silvestris* BL2 by growth substrates other than methane.

It was observed that the genes encoding NAR and NIR were expressed in strain T4 under suboxic (and anoxic) conditions without nitrogen oxyanions. It has similarly been reported in previous literature that exposure of *Paracoccus denitrificans* cells to N₂O under anoxia promoted NirS expression and synthesis⁴. We provide this information in the revised manuscript (SI, Lines 158-159). In the presence of nitrogen oxyanions, of course, we also expect the genes encoding NAR and NIR to be expressed in strain T4, but this result is not very informative.

The possibility of a 'potential batch effect' is low since we have more than three replicates and the data are always statistically consistent (*p*-value < 0.05). Nevertheless, as suggested by the reviewer, an exact mechanistic study of the induction and regulation of these genes is important. However, this will require further experimental and analytical

effort, along with the development of a genetic manipulation system for strain T4. We feel these detailed experiments are beyond the scope of the present paper, which is already at the upper limit of the journal's length constraints. The present study aims to document the growth of methanotrophs via N₂O reduction: more details on process regulation are important considerations for future study.

Ref: Please be aware that we moved the transcriptome data description of the original manuscript (Lines 451-470, 557-603) to SI (Lines 140-162, 185-233) due to the length constraint of this journal.

- In order to figure out why there was no growth of these methanotrophs when respiring on nitrate or nitrite, the authors carried out several experiments to link it to the accumulation of nitrite, and potential accumulation of C1 metabolites. I am wondering if the accumulation of NO could be another inhibitory factor. Checking the NO profile during the tests and even spike NO at different concentrations will not only clear up this point, but also shed light on the NO respiration of methanotrophs.

RE.) We agree with the reviewer's assessment that NO may be an inhibitory factor. We did mention this in the manuscript (original manuscript Lines 281-282; and revised manuscript Lines 277, 330-333). Revealing many aspects of NO function is a very interesting and key topic to understanding nitrogen cycling processes⁵. However, the technical difficulty of handling and measuring NO in a batch-culture system prevented us from pursuing this further. This can be avoided by using a chemostat-based system with a continuous headspace gas monitoring system instead. In future experiments, we intend to use a chemostat-based culture system to study how NO metabolism is linked to (anaerobic) growth of methanotrophs.

Minor comment

- There are some issues with references in SI, e.g., line 37, 189, 207.

RE.) Corrected (SI Lines 37, 319, 332-337)

Reviewer #3 (Remarks to the Author):

- This manuscript is very interesting and does give novel insights into the degree to which metabolic boundaries as we know them, can be stretched. I do think the notion that growth can be supported from N₂O as electron acceptor is very interesting. However, for the purposes of applications, I would encourage the authors to expand the sections where an application is promised or suggested. In particular where, can this be applied under the conditions methanotrophs work: methane rich conditions, and where only nitrous oxide is available to avoid nitrite toxicity? The experiments were heavily manipulated with customized conditions and electron donors that are not really what methanotrophs have in nature.

RE.) We appreciate the reviewer's comments and helpful suggestions. We have provided additional information on the potential ecological impact of these methanotrophs by including an additional paragraph titled "*Ecological relevance*" (line 528).

Some specific comments:

- **Line 47:** Authors should remove the application promise, even the core metabolisms of nitrate or nitrite dependent methane oxidation, have been promised since 2011 and still only a handful of pilot attempts with average performance have been achieved. Applications should not be oversold, and remain as ecological significance of the metabolism.

RE.) Removed as suggested.

- **Line 55:** the super index for references is not consistent, sometimes they have more spacing than others. Line 65.

RE.) Corrected throughout the manuscript.

- **Line 57:** methane is not estimated at 34 times, authors should update references.

RE.) Updated. (Line 57)

- **Line 99:** punctuation is odd, there needs to be a revision for ":"

RE.) Modified accordingly. (Line 99)

- **Line 211:** the term higher growth rates with higher values, means they grow slower. This nuance should be better evidenced in clearer writing.

RE.) The growth rate defined here is a measure of the maximum specific growth rates (μ_{max}). We have indicated this in the manuscript. Lines 204-206

- **Line 236:** the authors should include more details on the growth, these strains under the media used, are reported for the yield using DW measurements. What is this dry weight? Just bacterial cells? Is there any EPS produced? Is there any mass interference that can over estimate the yield? Methanotrophs are hardly pure and planktonic in all environments, and I am concerned with the yield relying on weight. Cell numbers with counting, or with

housekeeping 16S copy increase could have been better. In addition, FISH visual representation of the increase of specific % of few observations could evidence directly the yield truly comes from an increase in actual cells. Same applies for the growth, is based on OD values, and this is just an optical detection of light through a sample, but are cell numbers really increasing to the rates detected? Could the cells be storing and releasing carbon biopolymers, was the culture “flocculent” after some time? Was it really free-living planktonic cell units? I would appreciate, some insights into this.

RE.) Thank you for your comments. First, we would like to point out that the methanotrophic strains employed in this work are all pure isolates that appeared planktonic throughout the studies. Nonetheless, we addressed the reviewers' concerns by including data on cell counts and 16S rRNA copy increase in addition to OD₆₀₀ values. (Lines 209-211)

- **Line 260:** what is not clear to me at this point is, why base all on methanol? Methanol oxidation requires more energy in the cell as when starting with the intended goal of methane activation. I get that your experiments are not based on oxygen dependent activation, and that is why the conditions are altered from methanol to nitrous oxide as limits of the system. As valuable as it is that under these conditions, the growth from nitrous oxide as acceptor happens, the applicability or real occurrence may not be as significant as the results in vitro suggest.

RE.) As the reviewer commented, it is impossible to study the anaerobic growth of aerobic methanotrophs using methane since activation of methane requires O₂. Hence, methanol, commonly used by most methanotrophs as a growth substrate, was used in this study as ‘a model non-methane substrate’ since its oxidation does not require oxygen. Methanotrophs used in this study can be classified as habitat generalists due to their ability to reduce N₂O in addition to their flexibility in utilizable e-donors. Although our focus in this initial report was documenting the fundamentals of growth via N₂O reduction, other energy substrates used by these strains are important topics for future experiments.

Growth via N₂O reduction coupled to non-methane substrate oxidation might be more widespread in methanotrophs beyond the strains studied in our paper. *Methylocystis* spp. also commonly contain *nosZ* genes, and frequently utilize C-C compounds (such as ethanol and acetate)^{6, 7} and hydrogen as electron donors^{7, 8}, as exemplified in *Methylocystis* sp. strain SC2. This expands their niches to suboxic and anoxic environments, which is discussed in the new section of the revised manuscript. (Lines 554-557).

In addition, methanotrophs respiring N₂O might be able to grow at lower oxygen concentrations than those respiring oxygen only as the former ones can use the remaining traces of oxygen exclusively for methane activation. This was demonstrated for strain T4 in the current study (Lines 394-417) and could be an important growth strategy for N₂O reducing methanotrophs in natural environments. This is discussed in Lines 550-554.

Line 272: The growth dependent on the carbon fixing pathways that was observed under nitrous oxide, could then continue following the depletion of nitrate or nitrite reduction. This seems to not have been pursued, even under the same methanol conditions as energy source.

RE.) Although we are a bit confused about the reviewer's comments, we think the reviewer is implying that there might be two phases when *Methylocella* uses nitrate or nitrite as terminal electron acceptors. Initially, when methanol oxidation is combined with nitrate or nitrite reduction, toxic RNS and/or C1 intermediates inhibit cell growth. This was discussed in two sections of the original manuscript: '**Toxicity of reactive nitrogen species for *Methylocella* spp.**' and '**Toxicity of C1 metabolites in nitrate-reducing *Methylocella* spp.**' (Lines 274-333). Once nitrate and nitrite have been completely reduced, strains with complete denitrification pathway may further grow using N₂O accumulated during nitrate or nitrite reduction. However, this is not the case for incomplete denitrifiers. For example, in the case of strain T4, the absence of nitrite reductase prevents nitrate reduction from proceeding beyond nitrite (Lines 261-262). Strain BL2 cannot reduce the accumulated N₂O as it lacks NosZ (lines 262-264). Hence, for these strains, it would be impossible to pursue growth with nitrous oxide when nitrate or nitrite provided as the sole electron acceptor is depleted.

- **Line 304:** I would like to see more relation to real ecology, where will an environment where nitrite does not come from nitrate reduction and only nitrous oxide is the only available electron acceptor can be found?

RE.) Thank you for your comment. Following the suggestion, we added a new section titled '**Ecological relevance**' Lines 528-557. In terrestrial environments, various nitrogen oxides, originating from nitrification and denitrification processes, coexist and are spatiotemporally dynamic⁵. Thus, depending on the versatility of NO₂⁻ and NO reduction potential of methanotrophs as well as their coexistence with other NO₂⁻ and NO-reducing microorganisms, N₂O respiration can be supported or compromised (see revised manuscript Lines 330-333 and revised Supplementary information; Supplementary Figs. 5, 6). This will warrant further research in the context of natural environments.

- **Line 613:** if authors want to speculate on applications, this could be expanded as to how, under what conditions and provide insights so the field can follow this line of thought. Else, I would encourage the authors form suggesting applications in engineered systems.

RE.) In response to your comments, we removed the statements regarding direct application of these unique methanotrophs and focused on emphasizing the ecological importance of the newly discovered metabolic properties (lines 565-567).

- Work is reproducible, methodology is sound, and only the yield and growth estimations have some observations from my part for clarity. I end by stating that the experiments are diverse, complete, very interesting and overall, the work is relevant for a metabolic perspective, and for fundamental value.

RE.) We would like to thank the reviewer for the compliments and the excellent suggestions.

References

1. Buessecker S, et al. Coupled abiotic-biotic cycling of nitrous oxide in tropical peatlands. *Nat Ecol Evol* **6**, 1881-1890 (2022).
2. Mujakić I, et al. Multi-environment ecogenomics analysis of the cosmopolitan phylum *Gemmatimonadota*. *Microbiol Spectr* **11**, e0111223 (2023).
3. Bay SK, et al. Trace gas oxidizers are widespread and active members of soil microbial communities. *Nat Microbiol* **6**, 246-256 (2021).
4. Kellermann R, Hauge K, Tjåland R, Thalmann S, Bakken LR, Bergaust L. Preparation for denitrification and phenotypic diversification at the cusp of anoxia: a purpose for N₂O reductase vis-à-vis multiple roles of O₂. *Appl Environ Microbiol* **88**, e0105322 (2022).
5. Purchase ML, Bending GD, Mushinski RM. Spatiotemporal variations of soil reactive nitrogen oxide fluxes across the anthropogenic landscape. *Environ Sci Technol* **57**, 16348-16360 (2023).
6. Dedysh SN, Knief C, Dunfield PF. *Methylocella* species are facultatively methanotrophic. *J Bacteriol* **187**, 4665-4670 (2005).
7. Hakobyan A, Zhu J, Glatter T, Paczia N, Liesack W. Hydrogen utilization by *Methylocystis* sp. strain SC2 expands the known metabolic versatility of type IIa methanotrophs. *Metab Eng* **61**, 181-196 (2020).
8. Awala SI, et al. *Methylacidiphilum caldifontis* gen. nov., sp. nov., a thermoacidophilic methane-oxidizing bacterium from an acidic geothermal environment, and descriptions of the family *Methylacidiphilaceae* fam. nov. and order *Methylacidiphilales* ord. nov. *Int J Syst Evol Microbiol* **73**, (2023).